# Superposition as Lossy Compression – Measure with Sparse Autoencoders and Connect to Adversarial Vulnerability

**Leonard Bereska**[1]*, **Zoe Tzifa-Kratira**[1], **Reza Samavi**[2,3]†, **Efstratios Gavves**[1]
[1] *University of Amsterdam*
[2] *Toronto Metropolitan University*
[3] *Vector Institute for Artificial Intelligence*

**Reviewed on OpenReview:** *https://openreview.net/forum?id=qaNP6o5qvJ*

## Abstract

Neural networks achieve remarkable performance through *superposition*: encoding multiple features as overlapping directions in activation space rather than dedicating individual neurons to each feature. This phenomenon challenges interpretability: when neurons respond to multiple unrelated concepts, understanding network behavior becomes difficult. Yet despite its importance, we lack principled methods to measure superposition. We present an information-theoretic framework measuring a neural representation's *effective degrees of freedom*. We apply the Shannon entropy to sparse autoencoder activations to compute the number of effective features as the minimum number of neurons needed for interference-free encoding. Equivalently, this measures how many "virtual neurons" the network simulates through superposition. When networks encode more effective features than they have actual neurons, they must accept interference as the price of compression. Our metric strongly correlates with ground truth in toy models, detects minimal superposition in algorithmic tasks (effective features approximately equal neurons), and reveals systematic reduction under dropout. Layer-wise patterns of effective features mirror studies of intrinsic dimensionality on Pythia-70M. The metric also captures developmental dynamics, detecting sharp feature consolidation during the grokking phase transition. Surprisingly, adversarial training can increase effective features while improving robustness, contradicting the hypothesis that superposition causes vulnerability. Instead, the effect of adversarial training on superposition depends on task complexity and network capacity: simple tasks with ample capacity allow feature expansion (abundance regime), while complex tasks or limited capacity force feature reduction (scarcity regime). By defining superposition as lossy compression, this work enables principled, practical measurement of how neural networks organize information under computational constraints, in particular, connecting superposition to adversarial robustness.

## 1 Introduction

Interpretability and adversarial robustness could be two sides of the same coin (Räuker et al., 2023). Adversarially trained models learn more interpretable features (Engstrom et al., 2019; Ilyas et al., 2019), develop representations that transfer better (Salman et al., 2020), and align more closely with human perception (Santurkar et al., 2019). Conversely, interpretability-enhancing techniques improve robustness: input gradient regularization (Ross & Doshi-Velez, 2017; Boopathy et al., 2020), attribution smoothing (Etmann et al., 2019), and feature disentanglement (Augustin et al., 2020) all defend against adversarial attacks. Even architectural choices that promote interpretability, such as lateral inhibition (Eigen & Sadovnik, 2021) and second-order optimization (Tsiligkaridis & Roberts, 2020), yield more robust models. This pervasive duality demands mechanistic explanation.

---

*Corresponding author. Email: `leonard.bereska@gmail.com`
†Equal supervision

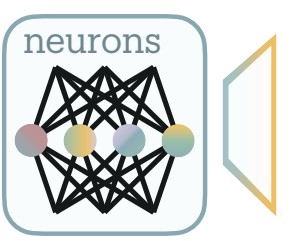
(a) Observed network

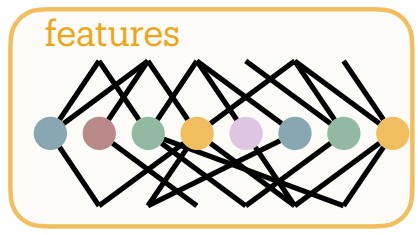
(b) Hypothetical disentangled model

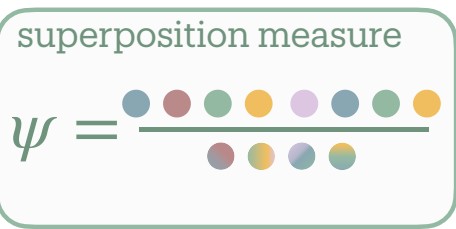
(c) Superposition as features per neuron

Figure 1: Defining superposition for a neural network layer. **(a)** Observed network with compressed representation where multiple features share neuronal dimensions. **(b)** Hypothetical disentangled model where each effective feature occupies its own neuron without interference (Elhage et al., 2022b). **(c)** Superposition measure $\psi$ quantifies effective features per neuron. Here, the network simulates twice as many effective features as it has neurons. Figure adapted from (Bereska & Gavves, 2024).

The superposition hypothesis offers a potential mechanism. Elhage et al. (2022b) showed that neural networks compress information through *superposition*: encoding multiple features as overlapping activation patterns. When features share dimensions, their interference creates attack surfaces that adversaries might exploit. If, by this mechanism, superposition caused adversarial vulnerability, this would explain *i.)* adversarial transferability as shared feature correlations (Liu et al., 2017), *ii.)* the robustness-accuracy trade-off as models sacrificing representational capacity for orthogonality (Tsipras et al., 2019), and *iii.)* robust models becoming more interpretable by reducing feature entanglement (Engstrom et al., 2019). Also, this superposition-vulnerability hypothesis predicts that *adversarial training should reduce superposition.*

Testing this prediction requires measuring superposition in real networks. While Elhage et al. (2022b) used weight matrix Frobenius norms, this approach requires ground truth features; available only in toy models. We need principled methods to quantify superposition without knowing the true features.

We solve this through information theory applied to sparse autoencoders (SAEs). SAEs extract interpretable features from neural activations (Cunningham et al., 2024; Bricken et al., 2023), decomposing them into sparse dictionary elements. We measure each feature's share of the network's representational budget through its activation magnitude across samples.

The exponential of the Shannon entropy quantifies how many interference-free channels would transmit this feature distribution, the network's *effective degrees of freedom.* We call this count *effective features* $F$ (Figure 1b): the minimum neurons needed to encode the observed features without interference. We interpret this as $F$ "virtual neurons": the network simulates this many independent channels through its $N$ physical neurons (Figure 1b). The feature distribution compresses losslessly down to exactly $F$ neurons; compress further and interference becomes unavoidable.

We measure superposition as $\psi = F/N$ (Figure 1c), counting virtual neurons per physical neuron. At $\psi = 1$, the network operates at its interference-free limit (no superposition). At $\psi = 2$, it simulates twice as many channels as it has neurons, achieving $2\times$ lossy compression. Thus, we define superposition as compression beyond the lossless limit.

Our findings contradict the simple superposition-vulnerability hypothesis. Adversarial training does not universally reduce superposition; its effect depends on task complexity relative to network capacity (Section 7). Simple tasks with ample capacity permit *abundance*: networks expand features for robustness. Complex tasks under constraints force *scarcity*: networks compress further, reducing features. This bifurcation holds across architectures (MLPs, CNNs, ResNet-18) and datasets (MNIST, Fashion-MNIST, CIFAR-10).

We validate the framework where superposition is observable. Toy models achieve $r = 0.94$ correlation through the SAE extraction pipeline (Section 5.1), and under SAE dictionary scaling the measure converges with appropriate regularization (Section 5.2). Beyond adversarial training, systematic measurement across contexts generates hypotheses about neural organization: dropout seems to act as capacity constraint, reducing superposition (Section 6.1), compressing networks trained on algorithmic tasks seems to not cre-

ate superposition ($\psi \leq 1$) likely due to lack of input sparsity (Section 6.2), during grokking, we capture the moment of algorithmic discovery through sharp drop in superposition at the generalization transition (Section 6.3), and Pythia-70M's layer-wise compression peaks in early MLPs before declining (Section 6.4); mirroring intrinsic dimensionality studies (Ansuini et al., 2019).

This work makes superposition measurable. By grounding neural compression in information theory, we enable quantitative study of how networks encode information under capacity constraints, potentially enabling systematic engineering of interpretable architectures.

## 2 Related Work

**Superposition and polysemanticity.** Neural networks employ distributed representations, encoding information across multiple units rather than in isolated neurons (Hinton, 1984; Olah, 2023). The discovery that semantic relationships manifest as directions in embedding space, exemplified by vector arithmetic like "king - man + woman = queen" (Mikolov et al., 2013), established the *linear representation hypothesis* (Park et al., 2023). Building on this geometric insight, Elhage et al. (2022b) formulated the *superposition hypothesis*: networks encode more features than dimensions by representing features as nearly orthogonal directions. Their toy models revealed phase transitions between monosemantic neurons (one feature per neuron) and polysemantic neurons (multiple features per neuron), governed by feature sparsity. Recent theoretical work proves networks can compute accurately despite the interference inherent in superposition (Vaintrob et al., 2024; Hänni et al., 2024).

While superposition (more effective features than neurons) inevitably creates polysemantic neurons through feature interference, polysemanticity (multiple features sharing a neuron) also emerges by other means: rotation of features relative to the neuron basis, incidentally (Lecomte et al., 2023) (e.g. via regularization), or forced by noise (such as dropout) as redundant encoding (Marshall & Kirchner, 2024) (as we show in Section 6.1, dropout shows the *opposite* effect on superposition). Scherlis et al. (2023) analyzed how features compete for limited neuronal capacity, showing that importance-weighted feature allocation can explain which features become polysemantic under resource constraints.

**Sparse autoencoders for feature extraction.** Sparse autoencoders (SAEs) tackle the challenge of extracting interpretable features from polysemantic representations by recasting it as sparse dictionary learning (Sharkey et al., 2022; Cunningham et al., 2024). SAEs decompose neural activations into sparse combinations of learned dictionary elements, effectively reversing the superposition process. Recent architectural innovations such as gated SAEs (Rajamanoharan et al., 2024), TopK variants (Gao et al., 2024; Bussmann et al., 2024), and Matryoshka SAEs (Bussmann et al., 2025) improve feature recovery. While our experiments employ vanilla SAEs for conceptual clarity, our entropy-based framework remains architecture-agnostic: improved feature extraction yields more accurate measurements without invalidating the theoretical foundation.

SAEs scale to state-of-the-art models: Anthropic extracted millions of interpretable features from Claude 3 Sonnet (Templeton et al., 2024), while OpenAI achieved similar results with GPT-4 (Gao et al., 2024). Crucially, these features are causally relevant: activation steering produces predictable behavioral changes (Marks et al., 2024). Applications span attention mechanism analysis (Kissane et al., 2024), reward model interpretation (Marks et al., 2023), and automated feature labeling (Paulo et al., 2024), establishing SAEs as foundational for mechanistic interpretability (Bereska & Gavves, 2024).

**Information theory and neural measurement.** Information-theoretic principles provide rigorous foundations for understanding neural representations. The information bottleneck principle (Tishby et al., 2000), when applied to deep learning (Shwartz-Ziv & Tishby, 2017), reveals how networks balance compression with prediction. Each neural layer acts as a bandwidth-limited channel, forcing networks to develop efficient codes (i.e. superposition) to transmit information forward (Goldfeld et al., 2019). This perspective recasts superposition as an optimal solution to rate-distortion constraints.

Most pertinent to our work, Ayonrinde et al. (2024) connected SAEs to minimum description length (MDL). By viewing SAE features as compression codes for neural activations, they showed that optimal SAEs

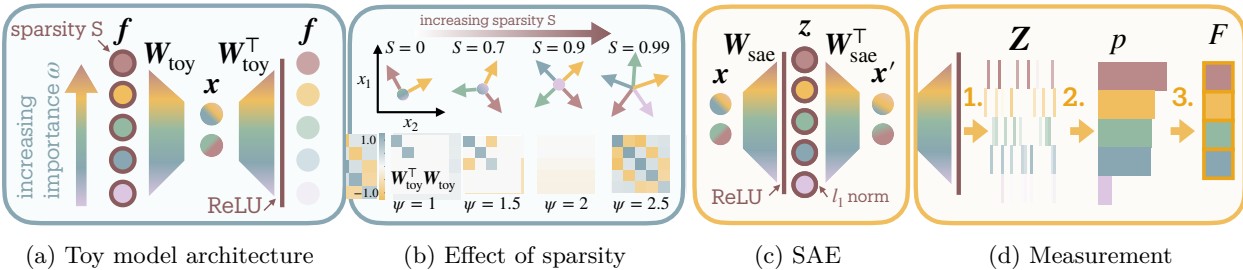

(a) Toy model architecture  (b) Effect of sparsity  (c) SAE  (d) Measurement

Figure 2: From toy model to practical superposition measurement. **(a)** Toy model bottlenecks features $\boldsymbol{f}$ through fewer neurons $\boldsymbol{x}$, with importance gradient determining allocation. **(b)** Sparsity enables interference-based compression: matrices $\boldsymbol{W}_{\text{toy}}^T \boldsymbol{W}_{\text{toy}}$ show off-diagonal terms growing as $\psi$ increases from 1 to 2.5. **(c)** Sparse autoencoders learn sparse codes $\boldsymbol{z}$ reconstructing activations $\boldsymbol{x}$. **(d)** Measurement: extract activations $\boldsymbol{Z}$, derive probabilities $p$, compute $F = e^{H(p)}$, measure $\psi = F/N$.

balance reconstruction fidelity against description complexity. Our entropy-based framework extends this perspective, measuring the effective "alphabet size" networks use for internal communication.

**Quantifying feature entanglement.** Despite its theoretical importance, measuring superposition remains unexplored. Elhage et al. (2022b) proposed a dimensions per feature metric for analyzing uniform importance settings in toy models, which when inverted could measure features per dimension. But this approach requires knowing the ground truth feature-to-neuron mapping matrix, limiting its applicability to controlled settings. Traditional disentanglement metrics from representation learning (Carbonneau et al., 2022; Eastwood & Williams, 2018) assess statistical independence rather than the representational compression characterizing superposition. Other dimensionality measures like effective rank (Roy & Vetterli, 2007) and participation ratio (Gao et al., 2017) quantify the number of significant dimensions in a representation but do not directly measure feature-to-neuron compression ratios.

Entropy-based measures have proven effective across disciplines facing similar measurement challenges. Neuroscience employs participation ratios (form of entropy, see Appendix A.3 for connection to Hill numbers) to quantify how many neurons contribute to population dynamics (Gao et al., 2017). Economics uses entropy to quantify portfolio concentration (Fontanari et al., 2021). Quantum physics applies von Neumann entropy to count effective pure states in entangled systems (Nielsen & Chuang, 2011). Recent work applies entropy measures to neural network analysis (Lee et al., 2023; Shin et al., 2024). Across fields, entropy naturally captures how information distributes across components: exactly what we need for measuring superposition.

## 3 Background on Superposition and Sparse Autoencoders

Neural networks must transmit information through layers with fixed dimensions. When neurons must encode information about many more features than available dimensions, networks employ *superposition*—packing multiple features into shared dimensions through interference. This compression mechanism enables representing more features than available neurons at the cost of introducing crosstalk between them. Superposition is compression beyond the lossless limit.

We examine toy models where superposition emerges under controlled bandwidth constraints, making interference patterns directly observable (Section 3.1). For real networks where ground truth remains unknown, we extract features through sparse autoencoders before measurement becomes possible (Section 3.2).

### 3.1 Observing Superposition in Toy Models

To understand how neural networks represent more features than they have dimensions, Elhage et al. (2022b) introduced minimal models demonstrating superposition under controlled conditions. The toy model com-

presses a feature vector $\boldsymbol{f} \in \mathbb{R}^M$ through a bottleneck $\boldsymbol{x} \in \mathbb{R}^N$ where $M > N$ (Figure 2a):

$$\boldsymbol{x} = \boldsymbol{W}_{\text{toy}}\boldsymbol{f},$$
$$\boldsymbol{f}' = \text{ReLU}(\boldsymbol{W}_{\text{toy}}^T\boldsymbol{x} + \boldsymbol{b}) \tag{1}$$

Here $M$ counts input features, $N$ counts bottleneck neurons, and $\boldsymbol{W}_{\text{toy}} \in \mathbb{R}^{N \times M}$ maps between them. The model must represent $M$ features using only $N$ dimensions; impossible unless features share neuronal resources.

Each input feature $f_i$ samples uniformly from $[0, 1]$ with sparsity $S$ (probability of being zero) and importance weight $\omega_i$. Training minimizes importance-weighted reconstruction error $\mathcal{L}(\boldsymbol{f}) = \sum_{i=1}^{M} \omega_i\|f_i - f_i'\|^2$, revealing how networks optimally allocate limited bandwidth.

As sparsity increases, the model packs features into shared dimensions through nearly-orthogonal arrangements (Figure 2b). The interference matrix $\boldsymbol{W}_{\text{toy}}^T\boldsymbol{W}_{\text{toy}}$ reveals this geometric solution: at low compression, strong diagonal with minimal off-diagonal terms; at high compression, substantial off-diagonal interference as features share space. These interference terms quantify the distortion networks accept for increased capacity. The ReLU nonlinearity proves essential, suppressing small interference to maintain reconstruction despite feature overlap.

Elhage et al. (2022b) proposed measuring "dimensions per feature" as $D^* = N/\|\boldsymbol{W}_{\text{toy}}\|_{\text{Frob}}^2$ for analyzing uniform importance settings, where the Frobenius norm $\|\boldsymbol{W}_{\text{toy}}\|_{\text{Frob}}^2 = \sum_{i,j} W_{ij}^2$ aggregates weight magnitudes. While this metric was not intended for general superposition measurement, we nevertheless adopt its inverse as a baseline, as it provides the only existing weight-based comparison point for our toy model validation:

$$\psi_{\text{Frob}} = \frac{\|\boldsymbol{W}_{\text{toy}}\|_{\text{Frob}}^2}{N} \tag{2}$$

This weight-based approach requires knowing the true feature-to-neuron mapping (unavailable in real networks) and lacks scale invariance (multiplying weights by any constant arbitrarily changes the measure). We need a principled framework quantifying compression *without* ground truth features.

## 3.2 Extracting Features Through Sparse Autoencoders

Real networks do not reveal their features directly. Instead, we must untangle them from distributed neural activations. Sparse autoencoders (SAEs) decompose activations into sparse combinations of learned dictionary elements, effectively reverse-engineering the toy model's feature representation (Cunningham et al., 2024; Bricken et al., 2023).

Given layer activations $\boldsymbol{x} \in \mathbb{R}^N$, an SAE learns a higher-dimensional sparse code $\boldsymbol{z} \in \mathbb{R}^D$ where $D > N$ (Figure 2c):

$$\boldsymbol{z} = \text{ReLU}(\boldsymbol{W}_{\text{enc}}\boldsymbol{x} + \boldsymbol{b}) \tag{3}$$

The reconstruction combines these sparse features:

$$\boldsymbol{x}' = \boldsymbol{W}_{\text{dec}}\boldsymbol{z} = \sum_{i=1}^{D} z_i\boldsymbol{d}_i \tag{4}$$

where columns $\boldsymbol{d}_i$ of $\boldsymbol{W}_{\text{dec}}$ form the learned dictionary.

Training balances faithful reconstruction against sparse activation:

$$\mathcal{L}(\boldsymbol{x}, \boldsymbol{z}) = \|\boldsymbol{x} - \boldsymbol{x}'\|_2^2 + \lambda\|\boldsymbol{z}\|_1 \tag{5}$$

The $\ell_1$ penalty creates explicit competition: the bound on total activation $\sum_i |z_i|$ forces features to justify their magnitude by contributing to reconstruction. This implements resource allocation where larger $|z_i|$ indicates greater consumption of the network's limited representational budget (see Appendix A.2 for rate-distortion derivation).

**SAE design choices.** We tie encoder and decoder weights ($\boldsymbol{W}_{\text{dec}} = \boldsymbol{W}_{\text{enc}}^T$) to enforce features as directions in activation space, maintaining conceptual clarity at potential cost to reconstruction (Bricken et al., 2023). Weight tying can also prevent feature absorption artifacts (Chanin et al., 2024a). We omit decoder bias following Cunningham et al. (2024) for a transparent baseline, accepting slight performance degradation. The $\ell_1$ regularization provides clean budget semantics, though alternatives like TopK (Gao et al., 2024) could work within our framework.

If networks truly employ superposition, SAEs should recover the underlying features enabling measurement. Recent work shows SAE features causally affect network behavior (Marks et al., 2024), suggesting they capture genuine computational structure. Our measurement framework remains architecture-agnostic: improved SAE variants enhance accuracy without invalidating the theoretical foundation.

## 4 Measuring Superposition Through Information Theory

We quantify superposition by determining how many neurons would be required to transmit the observed feature distribution without interference. Information theory provides a precise answer: Shannon's source coding theorem establishes that any distribution with entropy $H(p)$ can be losslessly compressed to $e^{H(p)}$ uniformly-allocated channels. This represents the minimum bandwidth for interference-free transmission—the network's *effective degrees of freedom.*

We formalize superposition as the compression ratio $\psi = F/N$, where $N$ counts physical neurons and $F = e^{H(p)}$ measures effective degrees of freedom extracted from SAE activation statistics (Figure 2d)[1]. When $\psi = 1$, the network operates at the lossless boundary. When $\psi > 1$, features necessarily share dimensions through interference. For instance, in Figure 2b, 5 features represented in 2 neurons yields $\psi = 2.5$.

**Feature probabilities from resource allocation.** Consider a layer with $N$ neurons whose activations have been processed by an SAE with dictionary size $D$. Across $S$ samples, the SAE produces sparse codes $\boldsymbol{Z} = \text{ReLU}(\boldsymbol{W}_{\text{sae}}\boldsymbol{X}) \in \mathbb{R}^{D \times S}$ where $\boldsymbol{X} \in \mathbb{R}^{N \times S}$ contains the original activations. Each feature's probability reflects its share of total activation magnitude[2]:

$$p_i = \frac{\sum_{s=1}^{S} |z_{i,s}|}{\sum_{j=1}^{D} \sum_{s=1}^{S} |z_{j,s}|} = \frac{\text{budget allocated to feature } i}{\text{total representational budget}} \tag{6}$$

The SAE's $\ell_1$ regularization ensures these allocations reflect computational importance. Features activating more frequently or strongly consume more capacity, with optimal $|z_i|$ proportional to marginal contribution to reconstruction quality (derivation in Appendix A.2).

**Effective features as lossless compression limit.** Shannon entropy quantifies the information content of this distribution: $H(p) = -\sum_i p_i \log p_i$. Its exponential:

$$F = e^{H(p)} \tag{7}$$

measures effective degrees of freedom, the minimum neurons needed to encode $p$ without interference. This is the network's lossless compression limit: the feature distribution could be transmitted through $F$ neurons with no information loss. Using fewer than $F$ neurons guarantees interference as features must share dimensions; using exactly $F$ achieves the interference-free boundary; the actual layer width $N$ determines whether compression remains lossless ($N \geq F$) or becomes lossy ($N < F$). The ratio

$$\psi = \frac{F}{N} \tag{8}$$

---

[1]In toy models where the input dimension $M$ is known, $F$ ranges from $N$ to $M$ depending on sparsity; in real networks $M$ is undefined and we estimate $F$ directly.

[2]Why not measure SAE weights instead of activations? Weight magnitude $\|\boldsymbol{w}_i\|$ indicates potential representation but misses actual usage: "dead features" may exist in the dictionary without ever activating. Empirically, a weight-based measure succeeds only in toy models (Figure 3a); and small toy transformer models already require our activation-based approach (Section 6.2).

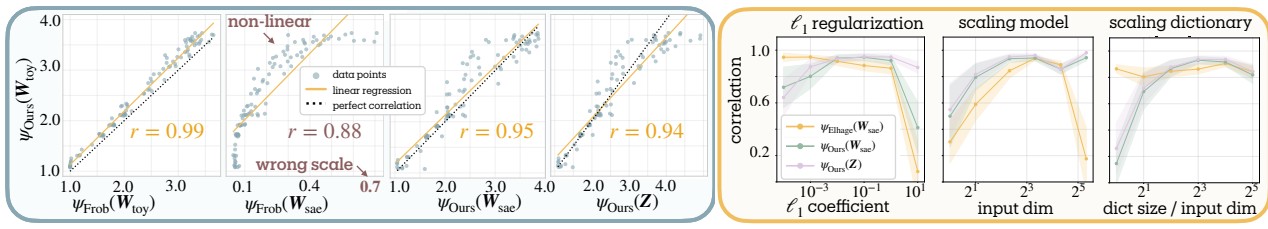

(a) Correlation with observable patterns     (b) Robustness across hyperparameters

Figure 3: Validation of superposition metrics. **(a)** Our measure maintains high correlation whether applied to toy weights ($r = 0.99$) or SAE activations ($r = 0.94$), while the Frobenius norm fails on SAE weights. **(b)** Performance remains stable across $\ell_1$ regularization, model scale, and dictionary size variations. Shaded regions show 95% confidence intervals across 100 model-SAE pairs.

then measures superposition as lossy compression.

While the SAE extracts $D$ *interpretable features*, semantic concepts humans might recognize, our measure quantifies $F$ *effective features*, the interference-free channel capacity required for their activation distribution. A network might use $D = 1000$ interpretable features but need only $F = 50$ effective features if most activate rarely.

Our measure inherits desirable properties from entropy. *i.)* For any $D$-component distribution, the output stays bounded $1 \leq F(p) \leq D$, bounded by single-feature dominance and uniform distribution. *ii.)* Unlike threshold-based counting, features contribute according to their information content: rare features matter less than common ones, weak features less than strong ones. This enables the interpretation as effective degrees of freedom, beyond "counting features".

In practice, we use sufficient samples until convergence (see convergence analysis in Section 6.4). For convolutional layers, we treat spatial positions as independent samples, measuring superposition across the channel dimension (Appendix A.7). While, in general, the data distribution for extracting SAE activations should reflect the training distribution, for studying adversarial training's effect, we evaluate on both clean inputs and adversarially perturbed inputs for contrast.

This framework enables quantifying superposition without ground truth by measuring each layer's compression ratio; how many virtual neurons it simulates relative to its physical dimension.

## 5 Validation of the Measurement Framework

### 5.1 Toy Model of Superposition

We validate our measure using the toy model of superposition (Elhage et al., 2022b), where interference patterns are directly observable. This controlled setting tests whether sparse autoencoders can recover accurate feature counts from superposed representations.

Following Elhage et al. (2022b), we generate 100 toy models with sparsity $S \in [0.001, 0.999]$. Each model compresses 20 features through a 5-neuron bottleneck, with importance weights decaying as $\omega_i = 0.7^i$. After training to convergence, we extract 10,000 activation samples and train SAEs with 40-dimensional dictionaries ($8\times$ expansion) and $\ell_1$ coefficient 0.1. This two-stage process mimics real-world measurement where ground truth remains unknown.

**Validation strategy.** Our validation proceeds in two steps. First, we establish reference values by measuring superposition directly from $\boldsymbol{W}_{\text{toy}}$, where the interference matrix $\boldsymbol{W}_{\text{toy}}^T \boldsymbol{W}_{\text{toy}}$ reveals compression levels: diagonal dominance indicates orthogonal features; off-diagonal terms show interference (Figure 2b). Both our entropy-based measure and the Frobenius norm baseline (Eq. 2) achieve near-perfect correlation ($r = 0.99 \pm 0.01$) when applied to toy model weights, confirming both track these observable patterns (Figure 3a).

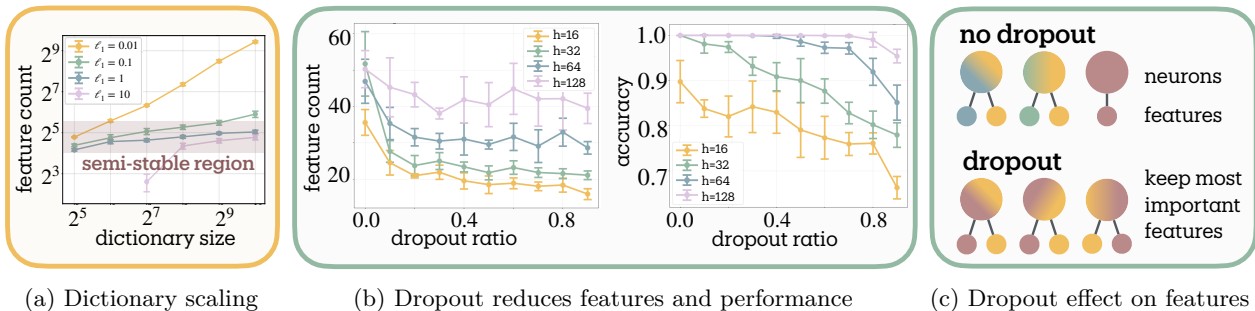

(a) Dictionary scaling      (b) Dropout reduces features and performance      (c) Dropout effect on features

Figure 4: Measurements on multi-task sparse parity dataset. **(a)** Dictionary scaling plateaus with proper regularization ($\ell_1 \geq 0.1$), validating intrinsic structure measurement. Weak regularization ($\ell_1 = 0.01$) shows unbounded growth through arbitrary subdivision. **(b)** Dropout monotonically reduces effective features and accuracy. **(c)** Capacity-dependent response: larger networks show reduced sensitivity while narrow networks exhibit sharp feature reduction, distinguishing polysemanticity (neurons encoding multiple features) from superposition (compression beyond lossless limit).

Second, we test whether each metric recovers these reference values when given only SAE outputs, the realistic scenario for measuring real networks. Here the Frobenius norm fails catastrophically on SAE weights, producing nonlinear relationships and incorrect scales (0.1–0.7 versus expected 1–4); the $\ell_1$ regularization fundamentally alters weight statistics. Our activation-based approach maintains strong correlation ($r = 0.94 \pm 0.02$) with the reference values even through the SAE bottleneck.

**Hyperparameter stability.** We test sensitivity across three axes: $\ell_1$ strength ($10^{-3}$ to $10^1$), model scale (8–32 input dimensions), and dictionary expansion ($2\times$ to $32\times$). Figure 3b shows stable performance across most configurations. Correlation degrades when extreme regularization ($\ell_1 = 10$) suppresses features, when dictionaries lack capacity to represent the feature set, when toy models are too small or too large to train reliably, or when very large dictionaries enable feature splitting (see Section 5.2). These failure modes reflect limitations of the toy model or SAE training rather than the measure itself.

## 5.2 Dictionary Scaling Convergence

Measuring a natural coastline with a finer ruler yields a longer measurement; potentially without bound (Mandelbrot, 1967). As SAE dictionaries grow, might we discover arbitrarily many features at finer scales?

We test convergence using multi-task sparse parity (Michaud et al., 2023) (3 tasks, 4 bits each) where ground truth bounds meaningful features. Networks with 64 hidden neurons trained across dictionary scales ($0.5\times$ to $16\times$ hidden dimension) and $\ell_1$ strengths (0.01 to 10.0).

Figure 4a reveals two regimes. With appropriate regularization ($\ell_1 \geq 0.1$), feature counts plateau despite dictionary expansion, indicating we measure the network's representational structure and not arbitrary decomposition (i.e. feature splitting (Chanin et al., 2024b)). Weak regularization ($\ell_1 = 0.01$) permits continued growth across all tested scales—this reflects feature splitting rather than genuine superposition, where the SAE decomposes single computational features into spurious fine-grained components. Excessive regularization ($\ell_1 = 10.0$) suppresses features entirely.

The dependence on dictionary size means absolute counts vary with SAE architecture, but comparative measurements remain valid: networks analyzed under identical configurations yield meaningful relative differences, even as changing those configurations shifts all measurements systematically.

# 6 Applications and Findings

We measure superposition across four neural compression phenomena: capacity constraint under dropout (Section 6.1), algorithmic tasks that resist superposition despite compression (Section 6.2), developmental

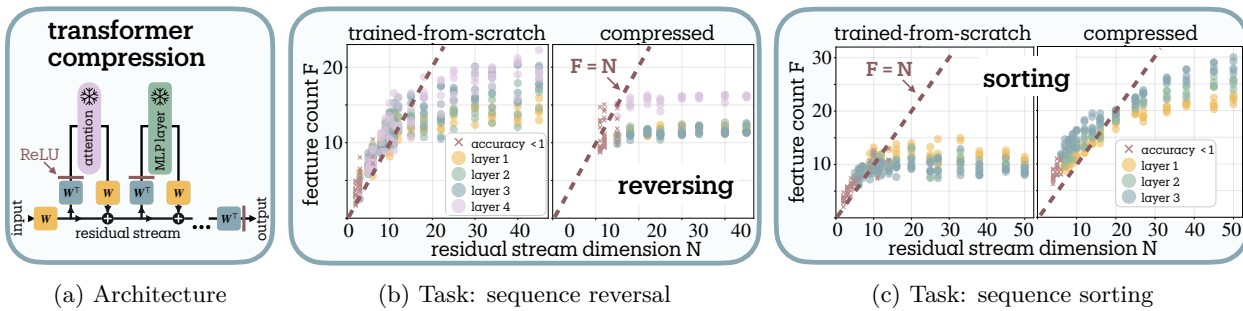

(a) Architecture      (b) Task: sequence reversal      (c) Task: sequence sorting

Figure 5: Algorithmic tasks under compression. **(a)** Compression architecture projects activations through $\mathrm{ReLU}(\boldsymbol{W}^\top \boldsymbol{W} \boldsymbol{x})$ to force features into fewer dimensions. **(b)** Sequence reversal: progressive compression from native 45D increases superposition from $\psi \approx 0.3$ toward $\psi = 1$ (leftmost to rightmost points approaching F=N line). Once reaching the F=N boundary, further compression causes performance degradation and then collapse ($\times$ markers) rather than superposition beyond $\psi = 1$. **(c)** Sorting exhibits identical dynamics with $F \approx N$ throughout compression. Both tasks resist genuine superposition ($\psi > 1$), operating at the lossless limit where each neuron encodes one effective feature, likely due to lack of input sparsity for sequence operations.

dynamics during learning transitions (Section 6.3), and layer-wise representational organization in language models (Section 6.4).

Each finding here is a preliminary, exploratory analysis on specific architectures and tasks. Our primary contribution remains the measurement tool itself. These findings illustrate its potential utility while generating testable hypotheses for future systematic investigation across broader experimental conditions.

## 6.1 Dropout Reduces Features Through Redundant Encoding

We investigate how dropout affects feature organization using multi-task sparse parity (3 tasks, 4 bits each) with simple MLPs across hidden dimensions $h \in \{16, 32, 64, 128\}$ and dropout rates $[0.0, 0.1, ..., 0.9]$.

Marshall & Kirchner (2024) showed dropout induces polysemanticity through redundancy: features must distribute across neurons to survive random deactivation. One might expect this redundancy to increase measured superposition. Instead, dropout monotonically reduces effective features by up to 50% (Figure 4b).

We propose this reflects the *distinction between polysemanticity and superposition* (Figure 4c). If dropout forces each feature to occupy multiple neurons for robustness, this redundant encoding would consume capacity, leaving room for fewer total features within the same dimensional budget. Under this interpretation, networks respond by pruning less essential features, consistent with Scherlis et al. (2023)'s competitive resource allocation framework.

The capacity dependence supports this account: larger networks show reduced dropout sensitivity while narrow networks exhibit sharp feature reduction, suggesting capacity constraints mediate the effect.

## 6.2 Algorithmic Tasks Resist Superposition Despite Compression

Tracr compiles human-readable programs into transformer weights with known computational structure (Lindner et al., 2023). We examine sequence reversal ("123" → "321") and sorting ("213" → "123"), comparing compiled models at their original dimensionality (compression factor 1×) against compressed variants and transformers trained from scratch with matching architectures.

Following Lindner et al. (2023), we compress models by projecting residual stream activations through learned compression matrices. Our compression scheme (Figure 5a) applies $\mathrm{ReLU}(\boldsymbol{W}^\top \boldsymbol{W} \boldsymbol{x})$ where $\boldsymbol{W} \in \mathbb{R}^{N \times M}$, compressing from originally $M$ dimensions to $N$. The ReLU activation, absent in the original Tracr compression, allows small interference terms to cancel out following the toy model rationale (Elhage et al., 2022b).

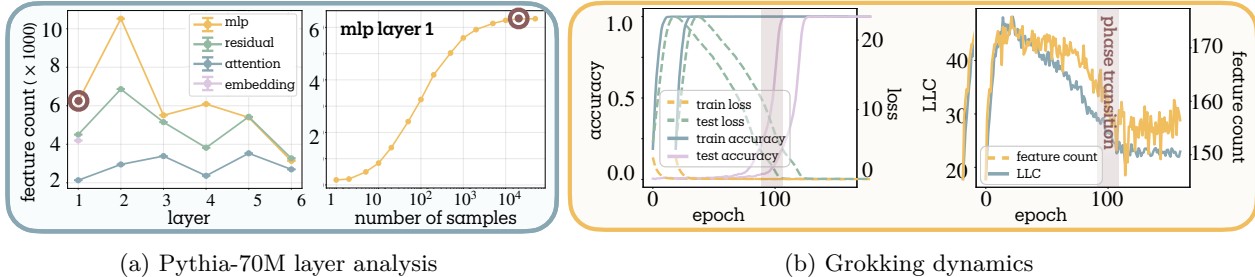

(a) Pythia-70M layer analysis        (b) Grokking dynamics

Figure 6: **(a)** Non-monotonic feature organization across Pythia-70M. MLP layer 1 peaks at 10,000 features ($20\times$ neurons). Convergence analysis shows saturation after $2 \times 10^4$ samples. **(b)** Feature dynamics during grokking on modular arithmetic. Sharp consolidation at generalization transition (epoch 60) follows smoother LLC decay. Strong correlation ($r = 0.908$, $p < 0.001$) with LLC suggests feature count functions as a measure of model complexity.

The compression dynamics reveal limits on superposition in these algorithmic tasks (Figure 5b, 5c). Both compiled Tracr models and transformers trained from scratch converge to 12 features for reversal and 10 for sorting[3]–far below their original compiled dimensions (45D for reversal), revealing substantial dimensional redundancy in Tracr's compilation.

As compression reduces dimensions from 45D toward the task-intrinsic boundary, superposition increases from $\psi \approx 0.3$ toward $\psi = 1$. However, compression stops increasing superposition once models reach the $F = N$ diagonal: further dimensional reduction causes linear drop in effective features and eventually performance collapse ($\times$ markers) rather than superposition beyond $\psi = 1$, resisting genuine superposition ($\psi > 1$) entirely.

This resistance likely stems from algorithmic tasks violating the sparsity assumption required for lossy compression (Elhage et al., 2022b). The toy model of superposition requires features to activate sparsely across inputs: most features remain inactive on most samples, keeping interference manageable. Algorithmic tasks break this assumption; sequence operations require consistent activation patterns across inputs. Without sparsity, interference becomes destructive rather than enabling compression. While we originally anticipated this setting would enable controlled validation across superposition levels, the systematic $F \approx N$ tracking, coupled with performance collapse when dimensions drop below this boundary, instead provides indirect evidence that our measure captures genuine capacity constraints, detecting minimal superposition as the sparsity prerequisite fails.

## 6.3 Capturing Grokking Phase Transition

Grokking (sudden perfect generalization after extended training on algorithmic tasks) provides an ideal testbed for developmental measurement (Power et al., 2022). We investigate whether feature count dynamics can detect this phase transition and how they relate to the Local Learning Coefficient (LLC) from singular learning theory (Hoogland et al., 2024).

We train a two-path MLP on modular arithmetic $(a + b)$ mod 53. Figure 6b reveals distinct dynamics: while LLC shows initial proliferation followed by smooth decay throughout training, our feature count exhibits sharp consolidation precisely at the generalization transition.

This pattern suggests the measures capture different aspects of complexity evolution. During memorization, the model employs numerous superposed features to store input-output mappings. The sharp consolidation coincides with algorithmic discovery, where the model reorganizes from distributed lookup tables into compact representations that capture the modular arithmetic rule (Nanda et al., 2023). Strong correlation

---

[3]While we generally recommend comparative interpretation due to measurement limitations (Section 8), the systematic $F = N$ boundary tracking and performance decline when violated suggest our measure may provide meaningful absolute effective feature counts in sufficiently constrained computational settings.

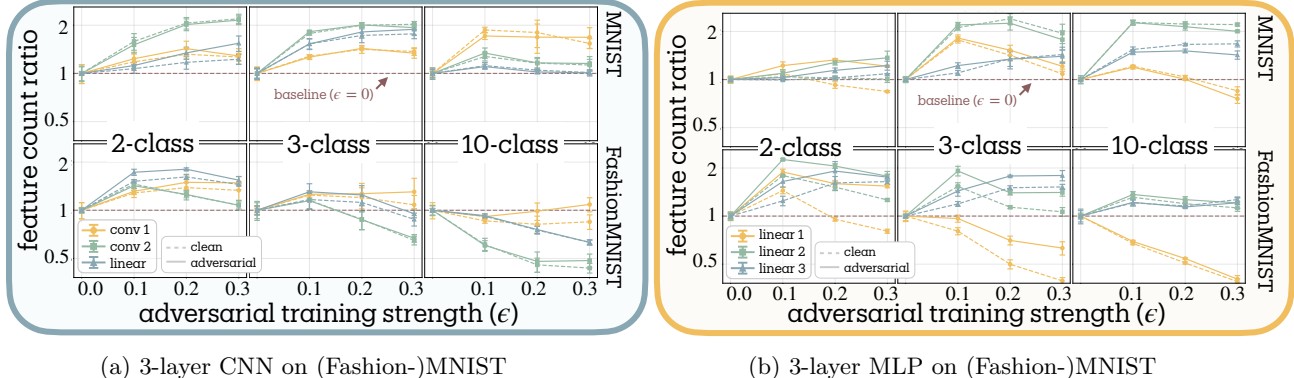

(a) 3-layer CNN on (Fashion-)MNIST         (b) 3-layer MLP on (Fashion-)MNIST

Figure 7: Higher task complexity shifts adversarial training's effect from feature expansion toward reduction. Each panel shows results varying dataset (MNIST top, Fashion-MNIST bottom) and number of classes (2, 3, 10). **(a)** CNNs show clear complexity-dependent transitions: simple tasks enable feature expansion while complex tasks (10 classes) force reduction below baseline. **(b)** MLPs exhibit similar patterns with more pronounced layer-wise variation. Fashion-MNIST consistently amplifies the reduction effect compared to MNIST, suggesting that representational demands drive defensive strategies beyond mere class count. Dashed lines: clean data; solid lines: adversarial examples. Feature count ratios normalized to $\epsilon = 0$ baseline. Error bars show standard error across 3 seeds.

($r = 0.908$, $p < 0.001$) between feature count and LLC positions superposition measurement as a developmental tool for detecting emergent capabilities through their information-theoretic signatures.

### 6.4 Layer-wise Organization in Language Models

We analyze Pythia-70M using pretrained SAEs from Marks et al. (2024), measuring feature counts across all layers and components. Convergence analysis (Figure 6a) shows saturation after $2 \times 10^4$ samples. Feature importance follows power-law distributions: while 21,000 SAE features activate for MLP 1, our entropy-based measure yields 5,600 effective features, automatically downweighting rare activations.

MLPs store the most features, followed by residual streams, with attention maintaining minimal counts, consistent with MLPs as knowledge stores and attention as routing (Geva et al., 2021). Features grow in early layers (MLP 1 achieves $20\times$ compression), compress through middle layers, then re-expand before final consolidation.

This non-monotonic trajectory parallels intrinsic dimensionality studies (Ansuini et al., 2019): both reveal "hunchback" patterns peaking in early-middle layers. Intrinsic dimensionality measures geometric manifold complexity (minimal dimensions describing activation structure), while we count effective information channels (minimal dimensions for lossless encoding), both measuring aspects of representational complexity.

## 7 Connection between Superposition and Adversarial Robustness

**Testing the superposition-vulnerability hypothesis.** The superposition-vulnerability hypothesis proposed by Elhage et al. (2022b) predicts that adversarial training should universally reduce superposition, as networks trade representational efficiency for orthogonal, robust features. We test this prediction systematically across diverse architectures and conditions, finding that the *direction* of the effect—expansion versus reduction—depends on task complexity and network capacity.

We employ PGD adversarial training (Madry et al., 2018) across architectures ranging from single-layer to deep networks (MLPs, CNNs, ResNet-18) on multiple datasets (MNIST, Fashion-MNIST, CIFAR-10). Task complexity varies through both classification granularity (2, 3, 5, 10 classes) and dataset difficulty. Network capacity varies through hidden dimensions (8–512 for MLPs), filter counts (8–64 for CNNs), and width scaling (1/4×–2× for ResNet-18). For convolutional networks, we measure superposition across channels

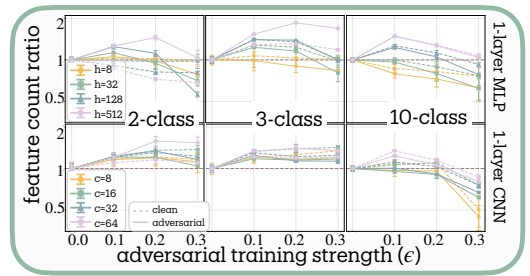 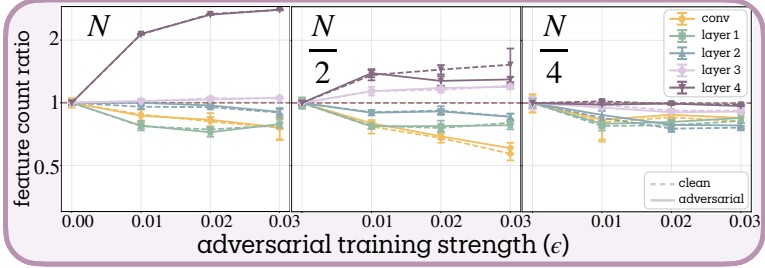

(a) Widening 1-layer NNs on MNIST                    (b) Narrowing ResNet-18 on CIFAR-10

Figure 8: Higher network capacity shifts adversarial training's effect from feature reduction toward expansion. **(a)** Single-layer networks on MNIST demonstrate capacity-dependent transitions: MLPs with hidden dimensions $h \in \{8, 32, 128, 512\}$ (top) and CNNs with filter counts $c \in \{8, 16, 32, 64\}$ (bottom) show that narrow networks reduce features while wide networks expand them across task complexities. **(b)** ResNet-18 on CIFAR-10 with width scaling ($1\times$, $1/2\times$, $1/4\times$) reveals layer-wise specialization: early layers reduce features while deeper layers (layer 3–4) expand dramatically, with this pattern dampening as width decreases. Dashed lines: clean data; solid lines: adversarial examples. Feature count ratios normalized to baseline.

by reshaping activation tensors to treat spatial positions as independent samples (see Appendix A.7 for details). All SAEs use $4\times$ dictionary expansion with $\ell_1 = 0.1$. Measurements on adversarial examples match the training distribution; models trained with $\epsilon = 0.2$ are evaluated on $\epsilon = 0.2$ attacks.

**Statistical methodology.** To quantify adversarial training effects, we extract normalized slopes representing how feature counts change per unit increase in adversarial training strength ($\epsilon \in \{0.0, 0.1, 0.2, 0.3\}$). Positive slopes indicate adversarial training *increases* features; negative slopes indicate *reduction*. For each experimental condition, we fit linear regressions to feature counts across epsilon values, pooling clean and adversarial observations to increase statistical power. These slopes are normalized by baseline ($\epsilon = 0$) feature counts, making effects comparable across layers with different absolute scales.

Since networks contain multiple layers, we aggregate layer-wise measurements using parameter-weighted averaging, where layers with more parameters receive proportionally higher weight. This reflects the assumption that computationally intensive layers better represent overall network behavior. For simple architectures, parameter counts include all weights and biases; for ResNet-18, we implement detailed counting that accounts for convolutions, batch normalization, and skip connections.

---

**Testing Adversarial Training Effects on Superposition**

We test three formal hypotheses:

- **H1 (Universal Reduction)**: Adversarial training uniformly reduces superposition across all conditions, directly testing Elhage et al. (2022b)'s original prediction.

- **H2 (Complexity ↓)**: Higher task complexity shifts adversarial training's effect from feature expansion toward reduction. We encode complexity ordinally (2-class=1, 3-class=2, 5-class=3, 10-class=4) and test for negative linear trends in the adversarial training slope.

- **H3 (Capacity ↑)**: Higher network capacity shifts adversarial training's effect from feature reduction toward expansion. We test for positive log-linear relationships between capacity measures and adversarial training slopes.

---

All statistical tests use inverse-variance weighting to account for measurement uncertainty, with random-effects meta-analysis when significant heterogeneity exists across conditions.

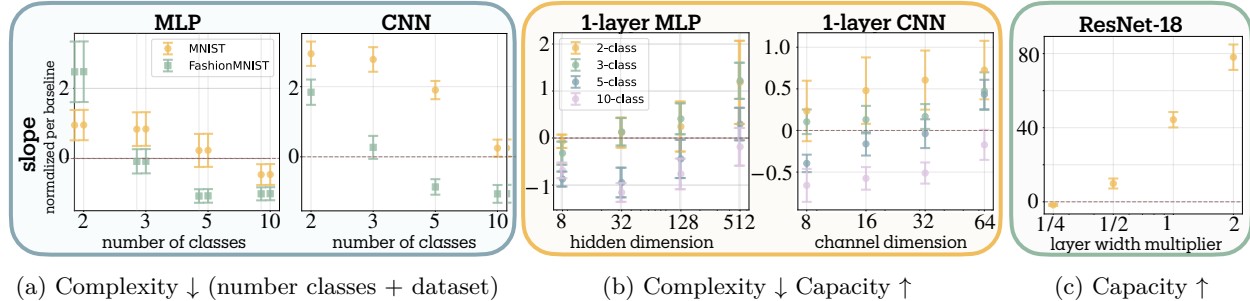

(a) Complexity ↓ (number classes + dataset)     (b) Complexity ↓ Capacity ↑     (c) Capacity ↑

Figure 9: Statistical analysis of adversarial training effects on superposition. Normalized slopes quantify feature count changes per unit adversarial strength $\epsilon$; positive slopes indicate adversarial training *increases* features, negative slopes indicate *reduction*. **(a)** Task complexity (number of classes + dataset difficulty) shows consistent negative relationship with slopes: higher complexity yields more negative slopes. Fashion-MNIST (green) produces systematically lower slopes than MNIST (yellow), consistent with its greater difficulty. **(b)** Single-layer networks on MNIST show capacity-dependent transitions: narrow networks (8–32 units) have negative slopes regardless of task complexity, while wide networks (128–512 units) have positive slopes. **(c)** ResNet-18 on CIFAR-10 demonstrates log-linear scaling: wider networks show dramatically more positive slopes. Error bars show standard errors.

**Complexity shifts adversarial training toward feature reduction (H2 supported).** Contrary to H1's prediction of universal reduction, adversarial training produces bidirectional effects whose direction depends systematically on task complexity (Figures 7 and 9a). Our meta-analysis reveals significant heterogeneity across conditions ($Q = 8.047$, $df = 3$, $p = 0.045$), necessitating random-effects modeling. The combined effect confirms H2: a negative relationship between task complexity and the adversarial training slope (slope $= -0.745 \pm 0.122$, $z = -6.14$, $p < 0.001$), meaning higher complexity shifts the effect from expansion toward reduction.

Binary classification consistently yields positive slopes, with feature counts expanding up to 2× baseline. Networks appear to develop additional defensive features when task demands are simple. Ten-class problems show negative slopes, with feature counts decreasing by up to 60%, particularly in early layers. Three-class tasks exhibit intermediate behavior with inverted-U curves: moderate adversarial training ($\epsilon = 0.1$) initially expands features before stronger training ($\epsilon = 0.3$) triggers reduction.

Dataset difficulty amplifies these effects. Fashion-MNIST produces systematically more negative slopes than MNIST (mean difference $= -1.467 \pm 0.156$, $t(7) = -2.405$, $p = 0.047$, Cohen's $d = -0.85$), consistent with its design as a more challenging benchmark (Xiao et al., 2017). This suggests that representational demands, beyond mere class count, drive defensive strategies.

Layer-wise patterns differ between architectures: MLP first layers reduce most while CNN second layers reduce most. We lack a mechanistic explanation for this divergence.

**Capacity shifts adversarial training toward feature expansion (H3 supported).** Network capacity exhibits a positive relationship with the adversarial training slope, strongly supporting H3 (Figures 8, 9b, and 9c). Single-layer networks demonstrate clear capacity thresholds (meta-analytic slope $= 0.220 \pm 0.037$, $z = 5.90$, $p < 0.001$). Networks with minimal capacity (8 hidden units for MLPs, 8 filters for CNNs) show negative slopes—reducing features across all task complexities—while high-capacity networks (512 units/64 filters) show positive slopes, expanding features even for 10-class problems.

This capacity dependence scales dramatically in deep architectures. ResNet-18 on CIFAR-10 exhibits a strong log-linear relationship between width multiplier and adversarial training slopes (slope $= 31.0 \pm 2.0$ per log(width), $t(2) = 15.7$, $p = 0.004$, $R^2 = 0.929$). An 8-fold width increase (0.25× to 2×) produces a 65-unit change in normalized slope. At minimal width (0.25×), adversarial training barely affects feature counts; at double width, networks show massive feature expansion with slopes approaching 80.

The layer-wise progression in ResNet-18 reveals hierarchical specialization: early layers (conv1, layer1) reduce features by up to 50%, middle layers remain stable, while deep layers (layer3, layer4) expand up to $4\times$. Systematically narrowing the network dampens this pattern: at 1/4 width, late-layer expansion vanishes while early-layer reduction persists but weakens. This could reflect vulnerability hierarchies, where early layers processing low-level statistics are easily exploited by imperceptible perturbations, necessitating feature reduction, while late layers encoding semantic information can safely expand their representational repertoire.

**Two regimes of adversarial response.** Our findings reveal a more nuanced relationship between superposition and adversarial vulnerability than originally theorized. Rather than universal feature reduction, adversarial training operates in two distinct regimes determined by the ratio of task demands to network capacity.

> **Bifurcation Driven by Task Complexity to Network Capacity Ratio**
>
> Adversarial training's effect on superposition depends on the ratio of task demands to network capacity:
>
> - **Abundance regime** (low complexity / high capacity): Adversarial training *increases* effective features. Networks add defensive features, achieving robustness through elaboration.
>
> - **Scarcity regime** (high complexity / low capacity): Adversarial training *decreases* effective features. Networks prune to fewer, potentially more orthogonal features, as predicted by the superposition-vulnerability hypothesis.

**Unexplained patterns.** Several patterns in our data remain unexplained. We observe non-monotonic inverted-U curves where moderate adversarial training ($\epsilon = 0.1$) expands features while stronger training ($\epsilon = 0.3$) reduces them below baseline. The gap between clean and adversarial feature counts varies unpredictably; sometimes negligible, sometimes substantial. Some results contradict our complexity hypothesis, with 2-class MLPs occasionally showing lower feature counts than 3-class. CNN experiments consistently yield stronger statistical significance ($p < 0.02$) than equivalent MLP experiments ($p \approx 0.09$) for unknown reasons.

**Implications for interpretability.** Our findings complicate simple accounts of why robust models often appear more interpretable (Engstrom et al., 2019). If interpretability benefits arose purely from reduced representational complexity, we would expect universal feature reduction under adversarial training. The existence of an abundance regime where feature counts *increase* suggests alternative mechanisms: perhaps non-interpretable shortcut features are replaced by richer, more human-aligned representations, or perhaps interpretability benefits are confined to the scarcity regime. Resolving this requires interpretability metrics beyond the scope of our current framework.

The bidirectional relationship between robustness and superposition suggests that achieving robustness without capability loss may require ensuring sufficient capacity for defensive elaboration. While our experiments demonstrate that increased robustness can coincide with either increased or decreased superposition depending on the regime, establishing the exact causal connection between superposition and robustness remains an important direction for future work.

## 8 Limitations

Our superposition measurement framework is limited by its dependence on sparse autoencoder quality, theoretical assumptions about neural feature representation, and should be interpreted as proxy for representational complexity rather than literal feature count:

**Sparse autoencoder quality.** Our approach inherently depends on sparse autoencoder feature extraction quality. While recent architectural advances (gated SAEs (Rajamanoharan et al., 2024), TopK variants

(Gao et al., 2024), and end-to-end training (Braun et al., 2024)) have substantially improved feature recovery, fundamental challenges remain. SAE training exhibits sensitivity to hyperparameters, particularly $\ell_1$ regularization strength and dictionary size, with different initialization or training procedures potentially yielding different feature counts for identical networks. Ghost features, i.e. SAE artifacts without computational relevance (Gao et al., 2024), can artificially inflate measurements, while poor reconstruction quality may deflate them.

**Assumptions on feature representation.** Our framework rests on several assumptions that real networks systematically violate. The linear representation assumption (that features correspond to directions in activation space) has been challenged by recent discoveries of circular feature organization for temporal concepts (Engels et al., 2024) and complex geometric structures beyond simple directions (Black et al., 2022). Our entropy calculation assumes features contribute independently to representation, but neural networks exhibit extensive feature correlations, synergistic information where feature combinations provide more information than individual contributions, and gating mechanisms where some features control others' activation. The approximation that sparse linear encoding captures true computational structure breaks down in hierarchical representations where low-level and high-level features are not substitutable, and in networks with substantial nonlinear feature interactions that cannot be decomposed additively.

**Comparative rather than absolute count.** Our measure quantifies effective representational diversity under specific assumptions rather than providing literal feature counts. This creates several interpretational limitations. The measure exhibits sensitivity to the activation distribution used for measurement. SAE training distributions must match the network's operational regime to avoid systematic bias. Feature granularity remains fundamentally ambiguous: broader features may decompose into specific ones in wider SAEs, creating uncertainty about whether we're discovering or creating features. Our single-layer analysis potentially misses features distributed across layers through residual connections or attention mechanisms. Most critically, we measure the effective alphabet size of the network's internal communication channel rather than counting distinct computational primitives, making comparative rather than absolute interpretation most appropriate.

The limitations largely reflect active research areas in sparse dictionary learning and mechanistic interpretability. Each advance in SAE architectures, training procedures, or theoretical understanding directly benefits measurement quality. Within its scope—comparative analysis of representational complexity under sparse linear encoding assumptions—the measure enables systematic investigation of neural information structure previously impossible.

## 9 Future Work

**Cross-model feature alignment.** Following Anthropic's crosscoder approach (Templeton et al., 2024), training joint SAEs across clean and adversarially-trained models would enable direct feature comparison. This could reveal whether the abundance regime involves feature elaboration (creating defensive variants) versus feature replacement (substituting vulnerable features with robust ones).

**Multi-scale and cross-layer measurement.** Current layer-wise analysis may miss features distributed across layers through residual connections. Matryoshka SAEs (Bussmann et al., 2025) already capture feature hierarchies at different granularities within single layers; extending this to cross-layer analysis could reveal how abstract features decompose into concrete features through network depth. Applying our entropy measure at each scale and depth would quantify information organization across both dimensions. Implementation requires developing new SAE architectures that span multiple layers.

**Feature co-occurrence and splitting.** Our independence assumption breaks when features consistently co-activate, yet this structure may be crucial for resolving feature splitting across dictionary scales. As we expand SAE dictionaries, single computational features can decompose into multiple SAE features - artificially inflating our count. Features that always co-occur likely represent such spurious decompositions rather than genuinely independent components. We initially attempted eigenvalue decomposition of

feature co-occurrence matrices to identify such dependencies, but this approach faces a fundamental rank constraint: covariance matrices have rank at most $N$ (the neuron count), making it impossible to detect superposition beyond the physical dimension. Alternative approaches include mutual information networks between features or hierarchical clustering of co-occurrence patterns. Combining these with Matryoshka SAEs' multi-scale dictionaries could reveal which features remain coupled across granularities (likely representing single computational primitives) versus those that split independently (likely representing distinct features). This would provide a principled solution to the dictionary scaling problem: count only features that disentangle across scales.

**Causal intervention experiments.** While we demonstrate correlation between adversarial training and superposition changes, establishing causality requires targeted interventions: *i.)* artificially constraining superposition via architectural modifications (e.g., softmax linear units (Elhage et al., 2022a)) then measuring robustness changes; *ii.)* directly manipulating feature sparsity in synthetic tasks; *iii.)* using mechanistic interpretability tools to trace how specific features contribute to adversarial vulnerability.

**Validation at scale.** Testing our framework on contemporary architectures (billion-parameter LLMs, Vision Transformers, diffusion models) would reveal whether findings generalize. Scale might expose new phenomena in adversarial training: very large models may escape capacity constraints entirely, or scaling laws might reveal limits on compression efficiency while maintaining robustness. If validated, our metric could guide architecture search for interpretable models by incorporating superposition measurement into training objectives or architecture design.

**Connection to model compression.** Our lossy compression perspective parallels findings in model compression research (Pavlitska et al., 2023). Both superposition (internal compression) and model compression (parameter reduction) force networks to optimize information encoding under constraints. Formalizing this connection through rate-distortion theory could yield theoretical bounds on the robustness-compression tradeoff, explaining when compression helps versus hurts.

## 10 Conclusion

This work provides a precise, measurable definition of superposition. Previous accounts characterized superposition qualitatively, as networks encoding "more features than neurons", we formalize it as *lossy compression*: encoding beyond the interference-free limit. Applying Shannon entropy to sparse autoencoder activations yields the effective degrees of freedom: the minimum neurons required for lossless transmission of the observed feature distribution. Superposition occurs when this count exceeds the layer's actual dimension.

The framework enables testing previously untestable hypotheses. The superposition-vulnerability hypothesis (Elhage et al., 2022b) predicts that adversarial training should universally reduce superposition as networks trade representational efficiency for orthogonality. We find instead that the effect depends on the ratio of task demands to network capacity: an *abundance* regime where simple tasks permit feature expansion, and a *scarcity* regime where complexity forces reduction. By grounding superposition in information theory, this work makes quantitative what was previously only demonstrable in toy settings.

## Author Contributions

**L.B.** conceived the project, developed the theoretical framework, designed and conducted all experiments except grokking, performed statistical analyses, and wrote the manuscript. **Z.T.-K.** conducted the grokking experiment and LLC comparison. **R.S.** and **E.G.** supervised the research.

## Acknowledgments

We thank Daniel Sadig for insightful discussions on adversarial training mechanisms and Hamed Karimi for detailed feedback on the manuscript that improved clarity and presentation. We are grateful to Jacqueline Bereska for valuable suggestions on manuscript organization and prioritization. We thank the anonymous TMLR reviewers for their rigorous feedback, particularly on statistical methodology and theoretical foundations, which substantially strengthened this work.

Part of this research was conducted during L.B.'s visit to the Trustworthy AI Lab (TAILab) at Toronto Metropolitan University. We are grateful for the stimulating research environment that facilitated the development of the core conceptual framework.

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

# A Theoretical Foundations

## A.1 Networks as Resource-Constrained Communication Channels

Neural networks must transmit information through layers with limited dimensions. Each layer acts as a communication bottleneck where multiple features compete for neuronal bandwidth. When a network needs to represent $F$ features using only $N < F$ dimensions, it uses lossy compression (:= superposition).

This resource scarcity creates a natural analogy to communication theory. Just as telecommunications systems multiplex multiple signals through shared channels, neural networks multiplex multiple features through shared dimensions. Our measurement framework formalizes this intuition by quantifying how efficiently networks allocate their limited representational budget across competing features.

## A.2 L1 Norm as Optimal Budget Allocation

The sparse autoencoder's $\ell_1$ regularization creates an explicit budget constraint on feature activations:

$$\mathcal{L}_{\text{SAE}} = \|\boldsymbol{h} - \boldsymbol{W}_{\text{sae}}^T \boldsymbol{z}\|_2^2 + \lambda \|\boldsymbol{z}\|_1 \tag{9}$$

The penalty term $\lambda\|\boldsymbol{z}\|_1 = \lambda \sum_i |z_i|$ enforces that the total activation budget $\sum_i |z_i|$ remains bounded. This creates competition where features must justify their budget allocation by contributing to reconstruction quality.

From the first-order optimality conditions of SAE training, the magnitude $|z_i|$ for any active feature satisfies:

$$|z_i| = \frac{1}{\lambda} |\boldsymbol{w}_i^T (\boldsymbol{h} - \boldsymbol{W}_{-i}^T \boldsymbol{z}_{-i})| \tag{10}$$

where $\boldsymbol{W}_{-i}$ excludes feature $i$. This reveals that $|z_i|$ measures the marginal contribution of feature $i$ to reconstruction quality—exactly the budget allocation that optimally balances reconstruction accuracy against sparsity. Our probability distribution therefore has meaning as "relative feature strength":

$$p_i = \frac{\mathbb{E}[|z_i|]}{\sum_j \mathbb{E}[|z_j|]} = \frac{\text{expected budget allocation to feature } i}{\text{total representational budget}} \tag{11}$$

This fraction represents how much of the network's limited representational resources are optimally allocated to feature $i$ under the SAE's constraints. Alternative norms fail to preserve this budget interpretation. The $\ell_2$ norm $\mathbb{E}[z_i^2]$ overweights outliers and breaks the linear connection to reconstruction contributions through squaring. The $\ell_\infty$ norm captures only peak activation while ignoring frequency of use. The $\ell_0$ norm provides binary active/inactive information but loses the magnitude data essential for measuring resource allocation intensity.

## A.3 Shannon Entropy as Information Capacity Measure

Given the budget allocation distribution $p$, the exponential of Shannon entropy provides the theoretically optimal feature count. The exponential of Shannon entropy, $\exp(H)$, is formally known as perplexity in information theory and the Hill number (order-1 diversity index) in ecology (Hill, 1973; Jost, 2006):

$$\text{PP}(p) = \exp\left(-\sum_i p_i \log p_i\right) = \prod_{i=1}^{n} p_i^{-p_i} \tag{12}$$

This quantifies the effective number of outcomes in a probability distribution: how many equally likely outcomes would yield identical uncertainty. In information theory, it represents the effective alphabet size of a communication system (Jelinek et al., 1977). In ecology, it quantifies the effective number of species in

an ecosystem (Jost, 2006). In statistical physics, it relates to the number of accessible states in a system (Jaynes, 1957). In quantum mechanics, it corresponds to the effective number of pure quantum states in a mixed state (Schrödinger, 1935).

Shannon entropy uniquely satisfies the mathematical properties required for principled feature counting (Anand et al., 2011). The measure exhibits coding optimality, equaling the minimum expected code length for optimal compression. It satisfies additivity for independent feature sets through $H(p \otimes q) = H(p) + H(q)$. Small changes in feature importance yield small changes in measured count through continuity. Uniform distributions where all features are equally important maximize the count. Adding features with positive probability monotonically increases the count. These axioms uniquely characterize Shannon entropy up to a multiplicative constant, making $\exp(H(p))$ the theoretically principled choice for aggregating feature importance into an effective count.

In quantum systems, von Neumann entropy $S(\rho) = -\mathrm{Tr}(\rho \log \rho)$ measures entanglement, with $e^{S(\rho)}$ representing effective pure states participating in a mixed quantum state (Nielsen & Chuang, 2011). Neural superposition exhibits parallel structure: just as quantum entanglement creates non-separable correlations that cannot be decomposed into independent subsystem states, neural superposition creates feature representations that cannot be cleanly separated into individual neuronal components. Both phenomena involve compressed encoding of information: quantum entanglement distributes correlations across subsystems resisting local description, while neural superposition distributes features across neurons resisting individual interpretation. Our measure $e^{H(p)}$ captures this compression by quantifying the effective number of features participating in the neural representation, analogous to how $e^{S(\rho)}$ quantifies effective pure states in an entangled quantum mixture.

Higher-order Hill numbers provide different sensitivities to rare versus common features:

$$^q\mathrm{D} = \left( \sum_{i=1}^{n} p_i^q \right)^{1/(1-q)} \tag{13}$$

where $q = 1$ gives our exponential entropy measure (via L'Hôpital's rule), $q = 0$ counts non-zero components, and $q = 2$ gives the inverse Simpson concentration index (participation ratio in statistical mechanics).

## A.4 Rate-Distortion Theoretical Foundation

Our measurement framework emerges from two nested rate-distortion problems that formalize the intuitive resource allocation perspective. The neural network layer itself solves:

$$R_{\mathrm{NN}}(D) = \min_{p(\boldsymbol{h}|\boldsymbol{x}):\mathbb{E}[d(\boldsymbol{y},f(\boldsymbol{h}))] \leq D} I(\boldsymbol{X};\boldsymbol{H}) \tag{14}$$

where the layer width $N$ constrains the mutual information $I(\boldsymbol{X};\boldsymbol{H})$ that can be transmitted, while $D$ represents acceptable task performance degradation. When the optimal solution requires representing $F > N$ features, superposition emerges naturally as the rate-optimal encoding strategy.

The sparse autoencoder solves a complementary problem:

$$R_{\mathrm{SAE}}(D) = \min_{p(\boldsymbol{z}|\boldsymbol{h}):\mathbb{E}[\|\boldsymbol{h}-\hat{\boldsymbol{h}}\|_2^2] \leq D} \mathbb{E}[\|\boldsymbol{z}\|_1] \tag{15}$$

where sparsity $\|\boldsymbol{z}\|_1$ acts as the rate constraint and reconstruction error as distortion. This dual structure justifies SAE-based measurement: we quantify the effective rate required to represent the network's compressed internal information under sparsity constraints.

The SAE optimization can be viewed as an information bottleneck problem balancing information preservation $\mathbb{E}[\|\boldsymbol{h} - g(\boldsymbol{z})\|_2^2]$ against information cost $\lambda \mathbb{E}[\|\boldsymbol{z}\|_1]$. Under this interpretation, $\mathbb{E}[|z_i|]$ represents the

information cost of including feature $i$ in the compressed representation, making our probability distribution a natural measure of information allocation across features.

### A.5 Critical Assumptions and Failure Modes

Our method measures effective representational diversity under sparse linear encoding, which approximates but does not exactly equal the number of distinct computational features. We must carefully assess the conditions under which this approximation holds.

**Feature Correspondence Assumption.** We assume SAE dictionary elements correspond one-to-one with genuine computational features. This assumption fails through feature splitting where one computational feature decomposes into multiple SAE features, artificially inflating counts. Feature merging combines multiple computational features into one SAE feature, deflating counts. Ghost features represent SAE artifacts without computational relevance (Gao et al., 2024). Incomplete coverage occurs when SAEs miss computationally relevant features entirely.

**Linear Representation Assumption.** We assume features combine primarily through linear superposition in activation space. Real networks violate this through hierarchical structure where low-level and high-level features aren't interchangeable. Gating mechanisms allow some features to control whether others activate (Elhage et al., 2022b). Combinatorial interactions emerge when meaning comes from feature combinations rather than individual contributions (Black et al., 2022).

**Magnitude-Importance Correspondence.** We assume $|z_i|$ reflects feature $i$'s computational importance. This breaks when SAE reconstruction preserves irrelevant details while missing computational essentials, when features interact nonlinearly in downstream processing (Engels et al., 2024), or when feature importance depends heavily on context rather than magnitude.

**Independent Information Assumption.** We assume Shannon entropy correctly aggregates information across features. This fails when correlated features don't contribute independent information, when synergistic information means feature pairs provide more information together than separately, or when redundant encoding has multiple features encoding identical computational factors.

The approximation captures genuine signal about representational complexity under specific conditions. The measure works best when features combine primarily through linear superposition, activation patterns are sparse with balanced importance, SAEs achieve high reconstruction quality on computationally relevant information, and representational structure is relatively flat rather than hierarchical. The approximation degrades with highly hierarchical representations, dense activation patterns with complex feature interactions, poor SAE reconstruction quality, or extreme feature importance skew. Despite these limitations, the measure provides principled approximation rather than exact counting, with primary value in comparative analysis across networks and training regimes.

### A.6 Why Eigenvalue Decomposition Fails for SAE Analysis

Following the quantum entanglement analogy, one might consider eigenvalue decomposition of the covariance matrix:

$$\mathbf{\Sigma} = \frac{1}{n} \boldsymbol{A}\boldsymbol{A}^T \tag{16}$$

where $\boldsymbol{A}$ represents the activation matrix. Eigenvalues $\{\lambda_1, \lambda_2, \ldots, \lambda_n\}$ represent explained variance along principal components, normalized to form a probability distribution:

$$p_i = \frac{\lambda_i}{\sum_{i=1}^{n} \lambda_i} \tag{17}$$

This approach faces fundamental rank deficiency when applied to SAEs. Expanding from lower dimension ($N$ neurons) to higher dimension ($D > N$ dictionary elements) yields covariance matrices with rank at most $N$, making detection of more than $N$ features impossible regardless of SAE capacity.

Our activation-based approach circumvents this limitation by directly measuring feature utilization through activation magnitude distributions rather than intrinsic dimensionality. This enables superposition quantification with overcomplete SAE dictionaries.

### A.7 Adaptation to Convolutional Networks

Convolutional neural networks organize features across channels rather than spatial locations. For CNN layers with activations $\mathcal{X} \in \mathbb{R}^{B \times C \times H \times W}$, we measure superposition across the channel dimension while accounting for spatial structure.

We extract features from each spatial location's channel vector independently, then aggregate when computing feature probabilities:

$$p_i = \frac{\sum_{b,h,w} |z_{b,i,h,w}|}{\sum_{j=1}^{D} \sum_{b,h,w} |z_{b,j,h,w}|} \tag{18}$$

where $z_{b,i,h,w}$ represents feature $i$'s activation at spatial position $(h, w)$ in sample $b$.

This aggregation treats the same semantic feature activating at different spatial locations (e.g., edge detectors firing everywhere) as evidence for a single feature's importance rather than separate features.

## B  Experimental Details

### B.1  Tracr Compression

We compile RASP programs using Tracr's standard pipeline with vocabulary $\{1, 2, 3, 4, 5\}$ and maximum sequence length 5. The sequence reversal program uses position-based indexing, while sorting employs Tracr's built-in sorting primitive with these parameters.

Following Lindner et al. (2023), we train compression matrices using a dual objective that ensures compressed models maintain both computational equivalence and representational fidelity:

$$\mathcal{L} = \lambda_{\text{out}} \mathcal{L}_{\text{out}} + \lambda_{\text{layer}} \mathcal{L}_{\text{layer}} \tag{19}$$

$$\mathcal{L}_{\text{out}} = \text{KL}(\text{softmax}(\mathbf{y}_c), \text{softmax}(\mathbf{y}_o)) \tag{20}$$

$$\mathcal{L}_{\text{layer}} = \frac{1}{L} \sum_{i=1}^{L} \|\mathbf{h}_i^{(o)} - \mathbf{h}_i^{(c)}\|_2^2 \tag{21}$$

where $\mathbf{y}_c$ and $\mathbf{y}_o$ denote compressed and original logits, and $\mathbf{h}_i^{(o)}$, $\mathbf{h}_i^{(c)}$ represent original and compressed activations at layer $i$.

Hyperparameters: $\lambda_{\text{out}} = 0.01$, $\lambda_{\text{layer}} = 1.0$, learning rate $10^{-3}$, temperature $\tau = 1.0$, maximum 500 epochs with early stopping at 100% accuracy. We use Adam optimization and train separate compression matrices for each trial. For each compressed model achieving perfect accuracy, we extract activations from all residual stream positions across 5 trials. SAEs use fixed dictionary size 100, L1 coefficient 0.1, learning rate $10^{-3}$, training for 300 epochs with batch size 128. We analyze the final layer activations (post-MLP) for consistency across compression factors.

### B.2  Multi-Task Sparse Parity Experiments

**Dataset Construction.**  We use the multi-task sparse parity dataset from Michaud et al. (2023) with 3 tasks and 4 bits per task. Each input consists of a 3-dimensional one-hot control vector concatenated with

12 data bits (total dimension 15). For each sample, the control vector specifies which task is active, and the label is computed as the parity (sum modulo 2) of the 4 bits corresponding to that task. This creates a dataset where ground truth bounds the number of meaningful features while maintaining task complexity.

**Model Architecture.** Simple MLPs with architecture Input(15) $\rightarrow$ Linear(h) $\rightarrow$ ReLU $\rightarrow$ Linear(1), where $h \in \{16, 32, 64, 128, 256\}$ for capacity experiments. We apply interventions (dropout) to hidden activations before the ReLU nonlinearity. Training uses Adam optimizer (lr=0.001), batch size 64, for 300 epochs with BCEWithLogitsLoss. Dataset split: 80% train, 20% test with stratification by task and label.

**Intervention Protocols. Dropout experiments**: Applied to hidden activations with rates [0.0, 0.1, 0.2, 0.3, 0.4, 0.5, 0.6, 0.7, 0.8, 0.9]. **Dictionary scaling**: Expansion factors [0.5, 1.0, 2.0, 4.0, 8.0, 16.0] relative to hidden dimension, with L1 coefficients [0.01, 0.1, 1.0, 10.0], maximum dictionary size capped at 1024. Each configuration tested across 5 random seeds with 3 SAE instances per configuration for stability measurement.

**SAE Architecture and Training.** Standard autoencoder with tied weights: $\mathbf{z} = \text{ReLU}(\mathbf{W}_{\text{enc}}\mathbf{x} + \mathbf{b})$, $\mathbf{x}' = \mathbf{W}_{\text{dec}}\mathbf{z}$ where $\mathbf{W}_{\text{dec}} = \mathbf{W}_{\text{enc}}^T$. Dictionary size typically 4$\times$ layer width unless specified otherwise.

Loss function: $\mathcal{L} = ||\mathbf{x} - \mathbf{x}'||_2^2 + \lambda||\mathbf{z}||_1$ with L1 coefficient $\lambda = 0.1$ (unless testing $\lambda$ sensitivity). Adam optimizer (lr=0.001), batch size 128, 300 epochs. For stability analysis, we train 3-5 SAE instances per configuration with different random seeds and report mean $\pm$ standard deviation.

## B.3 Grokking

**Task and Architecture.** Modular arithmetic task: $(a + b)$ mod 53 using sparse training data (40% of all possible pairs, 60% held out for testing). Model architecture: two-path MLP with shared embeddings.

$$\mathbf{e}_a = \text{Embedding}(a, \dim = 12) \tag{22}$$
$$\mathbf{e}_b = \text{Embedding}(b, \dim = 12) \tag{23}$$
$$\mathbf{h} = \text{GELU}(\mathbf{W}_1\mathbf{e}_a + \mathbf{W}_2\mathbf{e}_b) \tag{24}$$
$$\text{logits} = \mathbf{W}_3\mathbf{h} \tag{25}$$

where $\mathbf{W}_1, \mathbf{W}_2 \in \mathbb{R}^{48 \times 12}$ and $\mathbf{W}_3 \in \mathbb{R}^{53 \times 48}$.

**Training Configuration.** 25,000 training steps, learning rate 0.005, batch size 128, weight decay 0.0002. Model checkpoints saved every 250 steps (100 total checkpoints). Random seed 0 for reproducibility.

**LLC Estimation Protocol.** Local Learning Coefficient estimated using Stochastic Gradient Langevin Dynamics (SGLD) with hyperparameters: learning rate $3 \times 10^{-3}$, localization parameter $\gamma = 5.0$, effective inverse temperature $n_\beta = 2.0$, 500 MCMC samples across 2 independent chains. Hyperparameters selected via $5 \times 5$ grid search over epsilon range $[3 \times 10^{-5}, 3 \times 10^{-1}]$ ensuring $\varepsilon > 0.001$ for stability and $n_\beta < 100$ for $\beta$-independence.

## B.4 Pythia-70M Analysis

**Data Sampling and Preprocessing.** 20,000 samples from Pile dataset (Gao et al., 2020), shuffled with seeds [42, 123, 456] for reproducibility. Text preprocessing: truncate to 512 characters before tokenization to prevent memory issues. Tokenization using model's native tokenizer with `max_length=512`, `truncation=True`, no padding. Samples with empty text or tokenization failures excluded.

**Model and SAE Configuration.** Pythia-70M model with layer specifications: embedding layer, and $\{\text{attn\_out}, \text{mlp\_out}, \text{resid\_out}\}$ for layers 0–5. Pretrained SAEs from Marks et al. (2024) with dictionary size $64 \times 512 = 32,768$ features per layer. SAE weights loaded from subdirectories following pattern: `layer_type/10_32768/ae.pt`.

**Activation Processing.** Activations extracted using nnsight tracing with error handling for failed forward passes. Feature activations accumulated across all token positions and samples: feature_sum$_i$ = $\sum_{\text{samples,positions}} |\mathbf{z}_i|$. Feature count computed from accumulated sums using entropy-based measure. Memory management: explicit cleanup of activation tensors and CUDA cache clearing between seeds.

### B.5 Adversarial Robustness

#### B.5.1 Model Architectures

**Simple Models (Single Hidden Layer)**

- **SimpleMLP**: Input(784) $\to$ Linear($h$) $\to$ ReLU $\to$ Linear(output)
  - Hidden dimensions $h \in 8, 32, 128, 512$

- **SimpleCNN**: Input $\to$ Conv2d($h$, 5×5) $\to$ ReLU $\to$ MaxPool(2) $\to$ Linear(output)
  - Filter counts $h \in 8, 16, 32, 64$

**Standard Models**

- **StandardMLP**: Input(784) $\to$ Linear($4h$) $\to$ ReLU $\to$ Linear($2h$) $\to$ ReLU $\to$ Linear($h$) $\to$ ReLU $\to$ Linear(output)
  - Base dimension $h = 32$, yielding layer widths [128, 64, 32]

- **StandardCNN**: LeNet-style architecture
  - Conv2d(1, $h$, 3×3) $\to$ ReLU $\to$ MaxPool(2)
  - Conv2d($h$, $2h$, 3×3) $\to$ ReLU $\to$ MaxPool(2)
  - Linear($4h$) $\to$ ReLU $\to$ Linear(output)
  - Base dimension $h = 16$

**CIFAR-10 Models**

- **CIFAR10CNN**: Three-block CNN with batch normalization
  - Conv2d(3, $h$, 3×3) $\to$ BN $\to$ ReLU $\to$ MaxPool(2)
  - Conv2d($h$, $2h$, 3×3) $\to$ BN $\to$ ReLU $\to$ MaxPool(2)
  - Conv2d($2h$, $4h$, 3×3) $\to$ BN $\to$ ReLU $\to$ MaxPool(2)
  - Dropout(0.2) $\to$ Linear(output)
  - Base dimension $h = 32$

- **ResNet-18**: Modified for CIFAR-10
  - Initial: Conv2d(3, 64, 3×3, stride=1, padding=1)
  - MaxPool replaced with Identity
  - Standard ResNet-18 blocks [2, 2, 2, 2]

- **WideResNet**: ResNet-18 with variable width
  - Width factors: $1/16, 1/8, 1/4, 1/2, 1, 2, 4, 8$
  - Initial channels: $16 \times$ width factor
  - Block channels: $16, 32, 64, 128 \times$ width factor

### B.5.2 Training Protocols

**MNIST/Fashion-MNIST:**

- Optimizer: SGD with momentum 0.9

- Learning rate: 0.01, MultiStep decay at epochs [50, 75]

- Weight decay: $10^{-4}$

- Epochs: 100

- Batch size: 128

- PGD: 40 steps, step size $\alpha = 0.01$

- FGSM: Single step, $\alpha = \epsilon$

**CIFAR-10:**

- Optimizer: SGD with momentum 0.9

- Learning rate: 0.1, MultiStep decay at epochs [100, 150]

- Weight decay: $5 \times 10^{-4}$

- Epochs: 200

- Batch size: 128

- PGD: 10 steps, step size $\alpha = 2/255$

- FGSM: Single step, $\alpha = \epsilon$

### B.6 SAE Configuration

- Dictionary size: $4N$ ($4\times$ layer width)

- L1 coefficient: 0.1

- Optimizer: Adam, learning rate $10^{-3}$

- Training: 800 epochs with early stopping (patience 50)

- Activation collection: 10,000 samples from test set

- Separate SAEs trained per layer

