# OpenReview forum: "Superposition as Lossy Compression — Measure with Sparse Autoencoders and Connect to Adversarial Vulnerability"
_TMLR — Accepted by TMLR_

### Review · Reviewer_kgBT · 2025-08-18

**Summary Of Contributions:**

This paper introduces a new method to measure superposition—how neural networks encode multiple features into the same neurons. The authors use sparse autoencoders (SAEs) and an information theory formula (the exponential of Shannon entropy) to count the "effective number of features" in a network layer. The key finding contradicts the popular theory that superposition causes adversarial vulnerability. Instead of reducing superposition, the research shows that adversarial training often increases the number of features a model uses. The effect seems to be depending on the situation

**Audience:**

Yes

**Audience Explanation:**

In reviewing the paper, The submission's evidence is particularly strong because it is systematic and thorough. The authors don't just present a new metric; they rigorously validate it against known ground truths, test its stability, and then apply it to answer a significant open question in the field. I believe the audience will particularly find the main claim - relationship between superposition and adversarial robustness is more nuanced than previously thought—is especially compelling due to the breadth of experiments across different models, datasets, and conditions.

**Broader Impact Concerns:**

the work presented  is foundational and methodological, focusing on creating a tool to measure an internal property of neural networks. As such, it does not have direct ethical implications that would necessitate a detailed Broader Impact Statement.

**Claims And Evidence:**

Yes

**Claims Explanation:**

Yes, the claims made in this paper are supported by accurate, convincing, and clear evidence. It is a combination of theoretical grounding, validation in controlled settings, and large-scale empirical experiments—to build a strong case for both the authors proposed measurement technique and their subsequent scientific findings. In reviewing the paper, The submission's evidence is particularly strong because it is systematic and thorough. The authors don't just present a new metric; they rigorously validate it against known ground truths, test its stability, and then apply it to answer a significant open question in the field.

**Requested Changes:**

(1) Add Qualitative Examples of Feature Granularity - The paper notes that feature granularity is fundamentally ambiguous. I would recommend supplementing the quantitative dictionary scaling analysis (Figure 5a) with a qualitative example. Visualize a single, broad feature discovered by an SAE with a small dictionary and then show how it splits into multiple, more fine-grained features when a larger dictionary is used. This would make the abstract concept of granularity much more concrete.
(2) Propose a Method for Multi-Layer Analysis - The current analysis is performed on a per-layer basis. Including a "future work" section to outline a potential methodology for extending the analysis across layers or entire network blocks.
(3) Provide a experiment to Probe Causality - The paper demonstrates a strong correlation between adversarial training and changes in superposition but explicitly notes that establishing a direct causal link is an open question. It would benefit to strengthen the conclusion by proposing a concrete future experiment to test for causality.

---

> ### Author Response · Authors · 2025-09-25
> **Feature Granularity and Other Future Work**
>
> Thank you for your review and constructive suggestions. We appreciate your positive assessment of our theoretical framework and empirical validation.
>
> We appreciate this suggestion for visualizing feature splitting across dictionary scales. While we agree this would provide intuitive insight into the granularity phenomenon, we believe the quantitative convergence analysis in Figure 5a already demonstrates that with appropriate regularization, feature counts plateau despite dictionary expansion.
>
> Given timeline constraints and the fact that our primary contribution is the measurement framework itself (not the detailed phenomenology of SAE feature splitting, which has been explored in prior work like [[gao_scaling_2024]]), we propose to limit our scope to "*how do feature counts change*?" and leave "*how do specific features change*?" to future work (see cross-model feature alignment), which would clarify the underlying mechanisms especially driving our capacity×complexity observations.
>
> ### Future Work Section
>
> We absolutely agree that outlining future research directions would strengthen the paper. We will add a dedicated Future Work section:
>
> **Cross-model feature alignment.** Following Anthropic's crosscoder approach, training joint SAEs across clean and adversarially-trained models would enable direct feature comparison. This would reveal whether adversarial training creates new defensive features, transforms existing ones, or selectively prunes the feature repertoire. Such analysis could definitively answer whether the abundance regime involves feature elaboration versus replacement.
>
> **Multi-scale and hierarchical measurement.** Current layer-wise analysis may miss features distributed across layers through residual connections. While Matryoshka SAEs [[bussmann_learning_2025]] capture feature hierarchies at different granularities within a single layer, extending this to cross-layer analysis could reveal how abstract features in later layers decompose into concrete features in earlier layers. Combined with our entropy measure at each scale, this could quantify how information organization changes through network depth.
>
> **Causal mechanisms.** While we demonstrate correlation between adversarial training and superposition changes, establishing causality requires targeted interventions. Key experiments include: (1) artificially constraining superposition via architectural modifications (e.g., softmax linear units [[elhage_softmax_2022]]) then measuring robustness changes, (2) directly manipulating feature sparsity in synthetic tasks to test if reduced superposition improves adversarial robustness, (3) using mechanistic interpretability tools to trace how specific features contribute to adversarial vulnerability.
>
> **Feature co-occurrence and modularity.** Our independence assumption breaks down when features consistently co-activate. While eigenvalue decomposition of co-occurrence matrices faces rank constraints (limited by neuron count, preventing superposition measurement by construction), alternative approaches merit exploration. Mutual information between feature pairs or hierarchical clustering could identify feature modules that activate together. Combining this with Matryoshka SAEs' multi-scale analysis might reveal how feature dependencies organize across granularities—potentially explaining when the network treats features as independent versus interdependent components.
>
> **Connection to model compression.** Our lossy compression perspective parallels findings in model compression research, where similar paradoxes emerge regarding adversarial robustness [[pavlitska_relationship_2023]]. Just as we observe both expansion and reduction regimes for superposition under adversarial training, compressed models show contradictory effects on robustness—sometimes improving, sometimes degrading. This suggests a unified framework: both superposition (internal compression) and model compression (parameter reduction) force networks to optimize information encoding under constraints. Formalizing this connection through rate-distortion theory could yield theoretical bounds on the robustness-compression tradeoff, explaining when compression helps (forcing focus on robust features) versus hurts (losing defensive capacity).
>
> **Modern architectures.** Validating our framework on contemporary models (billion-parameter LLMs, Vision Transformers, diffusion models) would test whether our findings generalize. Scale might reveal new phenomena—perhaps very large models escape capacity constraints entirely, operating in permanent abundance regime where superposition becomes unnecessary.
>
> **Engineering interpretable systems.** If validated at scale, our metric could guide architecture search for interpretable models. By incorporating superposition measurement into training objectives or architecture design, we might develop models that naturally resist polysemanticity while maintaining performance.

---

### Review · Reviewer_5fcf · 2025-08-19

**Summary Of Contributions:**

**Summary**

The authors propose a novel metric for measuring superposition in neural network layers and use it to analyze several deep learning phenomena. The authors quantify superposition as the ratio of the effective number of features to the layer's dimensionality, where they calculate the effective number of features using a two-step procedure. During the first stage, the features encoded in a layer are disentangled with a sparse auto-encoder (SAE) trained over the layer's output. Then, the total number of features is quantified using an entropy-based metric calculated over the SAE's feature probabilities. Section 5 validates this metric on several problems (toy problem of superposition, Tractr, and sparse parity). Section 6 uses this metric to analyze several phenomena in deep learning: dropout, grokking, intrinsic dimensionality, and adversarial training. The authors specifically focus on adversarial training and show that this technique either increases the number of features in the abundance regime or decreases the number of features in the scarcity regime.


**Strengths**

1. The considered problem, the quantification of superposition, seems relevant for the interpretability of neural networks.
2. SAE seems to be an appropriate technique for feature disentanglement.
3. The choice of the entropy-based metric seems motivated by the related works.
4. Section 6 sheds light on the mechanism of some standard learning techniques. For instance, I think the conclusions of Sections 6.1 and 6.4 highlight some interesting underlying mechanisms of the considered phenomena.


**Weaknesses**

1. The metric is not well-motivated overall, even though I understand the logic behind some parts of its construction. Specifically, I do not understand what the effective number of features really measures (i.e., the semantics of this metric). For instance, I do not understand in which situations the entropy-based measure performs better than threshold-based measures (e.g., the dimensionality of SAE minus the number of "dead" neurons). Additionally, I do not think that the ratio of the number of features to the layer's dimensionality is well-motivated.
2. Additionally, I do not understand some choices made in Section 3.2. For instance, the authors omit the bias term in feature decoding and use weight tying for encoding and decoding matrices. These choices are not standard in the literature. Typically, bias terms are present during both encoding and decoding stages, and encoding and decoding matrices are optimized independently, even though they might share the initialization (Bricken et al., 2023, Appendix "Advice for Training Sparse Autoencoders: Autoencoder Architecture"). Moreover, the l1-regularization is often avoided due to the feature suppression problem (Wright and Sharkey, 2024; Rajamanoharan et al., 2024).
3. Section 5 does not sufficiently validate the metric. I will expand on this point in the following box.
4. The findings in Sections 6.1, 6.2, and 6.3 are not thoroughly tested. Each of these sections only considers one learning task and one architecture.
5. I think that the mechanisms proposed in Section 6.4 are not sufficiently validated. I will expand on this point in the following box.


**References**

Bricken, T., et al. (2023). Towards monosemanticity: Decomposing language models with dictionary learning.

Wright, B., and Sharkey, L. (2024). Addressing feature suppression in SAEs.

Rajamanoharan, S., et al. (2024). Improving sparse decomposition of language model activations with gated sparse autoencoders.

**Audience:**

Yes

**Audience Explanation:**

I think that the considered question is important for the interpretability of neural networks, and the mechanisms outlined in Sections 6.1 and 6.4 are interesting.

1. Superposition seems to be one of the main roadblocks to the interpretability of neural networks (e.g., in terms of neurons). Thus, metrics that quantify this phenomenon might be useful for research in this direction.
2. Section 6.1 suggests that dropout consumes the capacity of neural work, which refines the applicability of this technique.
3. Section 6.4 suggests that mechanisms of adversarial training might differ between small and big networks, which could potentially help with the analysis and application of this technique.
4. Overall, Section 6 demonstrated how the superposition metric might complement the analysis of several deep learning phenomena, which suggests that this metric is a valuable methodological tool.

**Broader Impact Concerns:**

I do not identify direct ethical implications of this work.

**Claims And Evidence:**

No

**Claims Explanation:**

Given the outlined weaknesses, I think that the article does not provide sufficient evidence for its claims.

1. I do not understand the scope of validation in Section 5. Specifically, I do not understand which properties of the metric are validated. Moreover, I think that the validity of the proposed tests is insufficient. Now, I will cover all subsections in more detail.
2. Section 5.1 compares the metric with the "weight mass" metric, allegedly proposed by Elhage et al. (2022). First, I want to note that Elhage et al. (2022) do not propose the "weight mass" metric. Elhage et al. (2022) only proposed a similar "dimensions per feature" metric, which they only used as a secondary metric in the analysis of uniform superposition (when all features have the same importance). However, the current manuscript uses this metric in the case of decaying feature importance, in which Elhage et al.'s (2022) metric does not semantically correspond to the ground truth. Moreover, the authors implicitly assume that $\psi_{ours}(W_{toy})$ constitutes a ground truth number of features in the toy setting, but they do not justify this claim.
3. Section 5.2 applies the new metric to Tractr-generated transformers. First, while the authors claim that they analyzed original transformers, I do not see these results in the submitted manuscript. Second, I do not understand the scope of validation in this setting. The authors only claim that feature counts stay below the $F=N$ line; this finding seems more fit for Section 6. (Also, note that Figure 4a is incorrect; Lindner et al. (2023) use transposed data and weight matrices, hence, all matrices in Figure 4a should also be transposed.)
4. Section 5.3 applies the new metric to sparse parity tasks and shows that feature count does not explode under an appropriate regularization. While I think that this finding strengthens the applicability of the authors' metric, I again do not understand what constitutes the ground truth for comparison. Specifically, I do not see how the authors conclude that they "we measure intrinsic structure, not artifacts" using their metric.
5. As I outlined above, I think that the scope of evaluation in Sections 6.1, 6.2, and 6.3 is insufficient.
6. As for Section 6.4, I think the current evaluation lacks rigor and mathematical precision. For example, in Figure 7 in the MNIST row, we can notice that the relative number of features of the first layer (linear 1 or conv 1) increases when we switch from 2-class classification to 3-class classification. Since 3-class classification is harder than 2-class classification, this observation contradicts the statement that "task complexity determines defensive strategy". Similarly, the behavior of conv, layer 1, and layer 3 is inconsistent in Figure 8b. I suggest that the authors use statistical tools to test these hypotheses to avoid confusion.

**Requested Changes:**

1. Please motivate the effective feature count metric in more detail.
2. Please explain the scope and the goal of experiments in Section 5.
3. For Section 5.1, please explain the motivation behind the "ground truth" measure or change the scope/positioning of evaluation.
4. For Sections 5.2, 6.1, 6.2, and 6.3, please either make the conclusions sound more speculative or extend the scope of experiments.
5. For Section 5.3, please explain the motivation in more detail and refine the scope of the conclusion.
6. For Section 6.4, please formalize the tested hypothesis and present the results of statistical testing.
7. Please motivate your design choices in Section 3.2.
8. (minor) Please fix Figure 4a and the corresponding description.
9. (minor) Please present findings for the original transformers in Section 5.2.
10. (minor) I think the paper would benefit from the discussion of effective rank (Roy and Vetterli, 2007) and participation ratio (Recanatesi et al., 2022) metrics, given that they approximate the rank of the matrix, which is close to the effective number of features.


**References**

Roy, O., and Vetterli, M. (2007). The effective rank: A measure of effective dimensionality.

Recanatesi, S., Bradde, S., Balasubramanian, V., Steinmetz, N. A., and Shea-Brown, E. (2022). A scale-dependent measure of system dimensionality.

---

> ### Author Response · Authors · 2025-09-24
> **1. Theoretical Motivation → Superposition as lossy compression**
>
> ## 1. Theoretical Motivation → Superposition as lossy compression
>
> Response to: *Please motivate the effective feature count metric in more detail.*
>
> Thank you for questioning our metric's theoretical motivation. Your concerns have prompted us to rethink and clarify what we believe now is a more suitable framing of superposition.
>
> > **TLDR;** in short, our measure quantifies superposition as the lossy compression factor neural networks employ to overcome capacity constraints. The effective feature count  represents the minimum neurons needed for lossless encoding of the observed feature statistics, while the ratio $\psi = F/N$ captures how much the network compresses beyond this ideal.
>
> We define **superposition as lossy compression**. To encode more information a neural network layer accepts some interference/crosstalk between features (the off-diagonal terms in  as shown in [[elhage_toy_2022]]) - this is the "distortion" in rate-distortion theory terms. We assume the compression achieved by the network to be optimal given the task (i.e. gradient descent to have worked), in particular, exhausting lossless compression before accepting interference.
>
> Our entropy-based measure $\hat{F} = e^{H(p)}$ quantifies **the number of neurons that would be sufficient to encode the observed feature distribution *without any interference***. We are measuring how many "*virtual neurons*" the network effectively simulates through superposition. If the network actually had $\hat{F}$ neurons available, it could represent all features without interference.
>
> Note that lossless compression is still possible regarding the SAE feature distribution (with many more features, e.g. the SAE latents - dead neurons) those can be compressed *without* loss (according to Shannon's coding theorem) to $\hat{F} = e^{H(p)}$ neurons (which then are uniformly distributed) → **any further compression is guaranteed to be lossy.**
>
> Also, note that this implies that even as we measure zero superposition, a network may still exhibit polysemanticity due to non-uniform feature statistics. We are not measuring compression in general, but only the lossy part. This demands that we **distinguish between our "virtual neurons" (effective features) and humanly meaningful features** that correspond to interpretable concepts - associated with the SAE latents. So, in particular, zero superposition does not entail a one-to-one neuron to humanly interpretable concept correspondence (as has been proposed as an alternative definition of what constitutes "solving superposition" [[elhage_toy_2022]]), but it entails no interference, which as we show in 6.4, seems relevant for the response to adversarial training. We will adjust our manuscript accordingly to make this clearer as it disentangles the different notions of superposition that exist in the literature.
>
> The ratio of effective features to neurons $\psi = \frac{F}{N}$ measures the **compression factor** achieved through superposition. When $\psi = 1$, the network uses its neurons without compression - one virtual neuron per physical neuron. When $\psi = 2.5$, the network achieves 2.5× lossy compression, simulating 2.5 virtual neurons for each physical neuron while accepting the inevitable interference.
>
> ##### No arbitrary thresholds
>
> For realistic feature distributions (often power law in language models), threshold-based counting suffers from extreme sensitivity. The derivative $dF/dt$ where $F$ is the count and $t$ is the threshold becomes arbitrarily large in power-law tails. A small threshold change can swing the count by orders of magnitude. Even setting the threshold to zero (only exclude dead neurons), just shifts the burden of hyperparameter-sensitive choice to the SAE size, scaling the number of SAE latents accordingly (especially with resampling dead neurons etc.).
> In contrast, the entropy measure is naturally somewhat robust to heavy tails: additional low-magnitude features contribute negligibly to $H(p)$ since their contribution scales as $p_i \log(1/p_i) \rightarrow 0$ for small $p_i  \rightarrow 0$.
>
> But more importantly, (and this is where our current manuscript is misleading/to-be-clarified given this improved theory) our **goal is not to count features, but effective degrees of freedom**. From a communication perspective, each layer transmits information forward with limited bandwidth. The exponential of entropy quantifies this effective **bandwidth** - how many equally-weighted channels would provide equivalent information capacity.
>
> We will revise the manuscript to foreground this (what we believe to be) clearer view. Again, thank you for eliciting this understanding.

---

> ### Author Response · Authors · 2025-09-24
> **2. Validation and 3. Ground Truth**
>
> ## 2. Validation Rationale
>
> Response to: *Please explain the scope and the goal of experiments in Section 5.*
>
> Section 5 validates specific measurement properties where ground truth or theoretical predictions are available.
>
> Properties we validate:
> - Correlation with observable ground truth (toy models): Tests whether our metric tracks known superposition levels in controlled settings where interference patterns are directly observable. Details are following (in 3. Ground Truth in Toy Model).
> - Robustness to hyperparameter variation (dictionary scaling, L1 strength): Ensures measurement stability across SAE training configurations.
> - Convergence properties (dictionary scaling): Validates that our metric captures intrinsic representational structure rather than artifacts of dictionary size under appropriate regularization.
>
> **Regarding Tracr Experiments (5.2):** We agree with the reviewer's assessment that these rather belong in applications (Section 6) rather than validation (Section 5). We originally intended to create controllable levels of superposition in Tracr models to validate our metric across known compression levels. However, we discovered that algorithmic tasks resist superposition entirely (F ≈ N), likely due to their lack of input sparsity—a requirement for the near-orthogonal feature arrangements that enable superposition in toy models.
>
> While this finding came as a surprise to us, it makes theoretical sense in hindsight: sequence operations require consistent activation patterns that prevent the sparse, near-orthogonal feature packing observed in toy models. The systematic tracking of the F = N boundary, coupled with performance collapse when violated, provides indirect evidence that our metric captures genuine capacity constraints as a measure of superposition/lossy compression should.
>
> However, for us this came as a discovery and may be validation only in hindsight. We will move it to Section 6 as application, but note that we do see it as indirect evidence of the quantitative (not just qualitative) validity of the "feature count" (→ zero sparsity = zero superposition).
>
> ## 3. Ground Truth in Toy Model
>
> Response to: *For Section 5.1, please explain the motivation behind the "ground truth" measure or change the scope/positioning of evaluation.*
>
> So, what is the ground truth in the toy model? We realize our manuscript was unclear on this point.
>
> > TLDR; In the toy model, **the ground truth is directly observable in the interference patterns** of $\mathbf{W}_{\text{toy}}^T\mathbf{W}_{\text{toy}}$.
>
> **Step 1: Visually observe ground truth in toy models**
> In [[elhage_toy_2022]]'s toy model (we used their "basic results" model with importance decay of $0.7$ that used for their visualizations, "to make it easy to visually see what happens." [[elhage_toy_2022]]), we **know** there are exactly 20 input features by construction. The model compresses these through 5 neurons. Then, the interference matrix directly reveals the number of input features actively encoded by the network, where diagonal elements  indicate actively represented features. As sparsity increases, we can **visually count** the transition from ~5 to ~20 represented features.
>
> Conceptually, we showed this in the bottom part of Figure 2b, but we will add a comprehensive appendix figure showing interference matrices for both $\mathbf{W}_{\text{toy}}^T\mathbf{W}_{\text{toy}}$ and $\mathbf{W}_{\text{sae}}\mathbf{W}_{\text{sae}}^T$  with the corresponding metric values.
>
> **Step 2: Validate metrics against observable patterns**
> Both the Frobenius measure and our entropy measure, when applied to $\mathbf{W}_{\text{toy}}$, produce values that track this visual progression. The **correlation between metrics is very high** ($r>0.99$) which validates that both metrics capture the **same underlying phenomenon** that we can directly observe in the interference patterns.
>
> **Step 3: Test robustness through SAE pipeline**
> The critical test now is: can we still measure superposition after extracting features with SAEs? Here, we take our metric on the toy model as ground truth, but since the correlation to the Frobenius measure is so high, these can be used interchangably).
>
> Also, thank you for identifying the attribution error. [[elhage_toy_2022]] proposed "dimensions per feature" $D^* = m/|\mathbf{W}|_F^2$ specifically for uniform importance settings, not the inverse measure $|\mathbf{W}|_F^2/m$ we used. We will correct this and rename our baseline to **"Frobenius norm measure"** to avoid misattribution. While this metric wasn't formally proposed by Elhage et al., we see it as a natural weight-based approach to measure superposition.
>
> We will revise Section 5.1 to make this validation logic explicit, and revising the sentiment: so that we're not claiming superiority over a metric that was never proposed for this purpose, but demonstrate that our approach solves the measurement problem where ground truth is unavailable.

---

> ### Author Response · Authors · 2025-09-24
> **4. Findings more speculative 5. Dictionary Scaling Motivation and 8. + 9. Minor**
>
> ## 4. State Findings More Speculative
>
> Response to: _For Sections 5.2, 6.1, 6.2, and 6.3, please either make the conclusions sound more speculative or extend the scope of experiments._
>
> We agree. These findings represent exploratory applications demonstrating measurement utility across diverse contexts rather than definitive claims. Our primary contribution is the measurement tool itself—each application deserves dedicated investigation with appropriate statistical power.
>
> We will revise these sections to position findings as preliminary observations that generate hypotheses for future work.
>
>
> ## 5. Dictionary Scaling Motivation and Claims
>
> Response to: _For Section 5.3, please explain the motivation in more detail and refine the scope of the conclusion._
>
> The dictionary scaling experiment tests whether our metric reflects the network's representational economy or just the SAE's capacity to subdivide information. If enlarging the dictionary automatically increases measured features, we would be measuring artifacts of the measurement tool and not network properties.
>
> Under sufficient regularization pressure, feature counts plateau despite dictionary expansion. This suggests our metric does not yield to arbitrary decomposition.
>
> The phrase "we measure intrinsic structure, not artifacts" is indeed poorly defined and the "intrinsic structure" part unsupported by these specific results. This has escaped our proof-reading. We will clarify the writing.
>
> ## 8. (minor) Transpose matrices
>
> Response to: *Please fix Figure 4a and the corresponding description.*
>
> Thank you for catching the matrix orientation error. Lindner et al. (2023) indeed use transposed conventions.
>
> ## 9. (minor) Original Transformers
>
> Response to: *Please present findings for the original transformers in Section 5.2.*
>
> The "original transformers" corresponds to compression factor 1× in Figure 4b,c
> as the leftmost data points. We will make this explicit in the caption and clarify in the section.

---

> ### Author Response · Authors · 2025-09-24
> **6. Formalizing Hypotheses and Statistical Testing**
>
> ## 6. Formalizing Hypotheses and Statistical Testing
>
> *For Section 6.4, please formalize the tested hypothesis and present the results of statistical testing.*
>
> We appreciate the reviewer's detailed examination of our adversarial robustness findings and agree that our initial descriptive approach lacked statistical rigor. The apparent inconsistencies you identified—such as the first layer's behavior between 2-class and 3-class MNIST—highlight why formal hypothesis testing was essential. We have completely revised Section 6.4 with a comprehensive statistical framework and invite you specifically to inspect Figure 9.
>
> ##### Formalized Hypotheses
>
> We now test three explicit hypotheses:
>
> **H1 (Universal Reduction Hypothesis)**: Following [[elhage_toy_2022]]'s superposition-vulnerability theory, adversarial training should universally reduce superposition across all conditions as networks trade representational efficiency for orthogonal, robust features.
>
> **H2 (Task Complexity Modulation)**: Task complexity systematically modulates adversarial training effects. We encode complexity ordinally (2-class=1, 3-class=2, 5-class=3, 10-class=4) and test for linear trends in feature count responses.
>
> **H3 (Capacity Enabling Hypothesis)**: Network capacity enables different defensive strategies. Higher capacity permits feature expansion while constrained networks force reduction.
>
> ##### Statistical Methodology
>
> We developed a multi-level statistical framework extracting normalized slopes representing feature count changes per unit adversarial strength ($\epsilon$), using parameter-weighted aggregation across layers with inverse-variance weighting to handle measurement uncertainty.
>
> ##### Key Statistical Findings
>
> **Against Universal Reduction (H1)**: Meta-analysis reveals significant heterogeneity ($Q = 8.047$, $df = 3$, $p = 0.045$). The random-effects model yields an overall negative trend (slope = $-0.745 \pm 0.122$, $z = -6.14$, $p < 0.001$), but individual conditions show bidirectional effects, decisively rejecting the universal reduction hypothesis.
>
> **Task Complexity Effects (H2)**: CNN architectures show significant negative trends (MNIST: slope = -0.953 ± 0.133, $p = 0.019$; Fashion-MNIST: slope = -0.931 ± 0.136, $p = 0.021$), while MLP results are borderline ($p \approx 0.09$). Our random-effects meta-analysis yields a highly significant combined effect (slope = $-0.745 \pm 0.122$, $z = -6.14$, $p < 0.001$), demonstrating that while individual experiments may be underpowered, the systematic pattern strongly supports complexity modulation.
>
> **Capacity Effects (H3)**: Single-layer networks demonstrate significant capacity-dependent responses (meta-analytic slope = $0.220 \pm 0.037$, $z = 5.90$, $p < 0.001$). The effect scales dramatically in deep networks—ResNet-18 shows log-linear scaling with width (slope = $31.0 \pm 2.0$ per $\log(\text{width})$, $t(2) = 15.7$, $p = 0.004$, $R^2 = 0.929$).
>
> ##### Addressing Specific Inconsistencies
>
> The reviewer correctly identifies several patterns we cannot fully explain. The MLP first-layer behavior between 2-class and 3-class MNIST contradicts our ordinal complexity encoding, suggesting discontinuous transitions we don't capture. We observe non-monotonic responses where moderate adversarial training expands features before stronger training causes reduction, and opposing layer-wise patterns between MLPs and CNNs that likely reflect architectural differences we don't yet understand.
>
> We acknowledge these limitations explicitly in our revision and thank the reviewer for ensuring we clearly delineate established findings from unexplained phenomena.
>
> ##### Statistical Support for Theoretical Framework
>
> These statistical findings support the two-regime model of adversarial robustness: **abundance regime** (low complexity, high capacity) enables feature expansion while **scarcity regime** (high complexity, low capacity) forces consolidation. This framework, now grounded in rigorous statistical testing, explains the systematic patterns across our experimental matrix.

---

> ### Author Response · Authors · 2025-09-24
> **7. SAE Design Choices**
>
> ## 7. SAE Design Choices
>
> Response to: *Please motivate your design choices in Section 3.2.*
>
> > **TLDR:** Our SAE choices prioritize **simplicity** and **conceptual clarity** over state-of-the-art performance. The entropy framework remains architecture-agnostic—better feature extraction yields more accurate measurements without invalidating the theory.
>
> We agree that **decoder bias** [[bricken_monosemanticity_2023]] likely helps with minimal downside. Instead we followed [[cunningham_sparse_2024]] for a simpler setup. We cannot retroactively modify experiments but will acknowledge this limitation.
>
> **Weight tying ($\mathbf{W}_{\text{dec}} = \mathbf{W}_{\text{enc}}^T$)** enforces the geometric interpretation of features as directions in activation space. We chose this **conceptual clarity**, over reconstruction performance. There is some evidence in toy models that weight tying can **prevent feature absorption** [[chanind_toy_2024]], but agreed - most current practice favors the less restrictive untied weights [[bricken_monosemanticity_2023]].
>
> **L1 regularization** encourages features to compete for limited representational budget:
>
> $$p_i = \frac{\mathbb{E}[|z_i|]}{\sum_j \mathbb{E}[|z_j|]} = \frac{\text{optimal budget allocation to feature } i}{\text{total representational budget}}$$
> This budget framework provides clean theoretical motivation—features with larger allocations (activation magnitude times frequency) contribute more to reconstruction. However, our entropy-based framework could operate on any activation distribution. Alternative regularizers (L2, TopK) would yield different activation patterns but the exponential entropy $e^{H(p)}$ would still measure effective feature count; yet, not based on magnitude, but on frequency alone.
> We referenced to the appendix, but will make this more explicit in the main part.
>
> Overall, we followed [[cunningham_sparse_2024]] (except for the weight-tying) for a conceptually transparent ("vanilla") baseline rather than chasing architectural SoTA. We would be excited about future work (e.g. Matryoshka SAEs for more principled treatment of feature hierarchy, and TopK, or gated variants etc.) to improve upon our intentionally simple setup. However, these advances would enhance measurement accuracy *within our measurement framework*.
>
> We will clarify Section 3.2 to better motivate these choices while acknowledging our architectural trade-offs. The core contribution—entropy-based superposition measurement—remains robust across architectural variants.

---

> ### Comment · Reviewer_5fcf · 2025-09-26
> **Response the authors' comments**
>
> Thanks for the detailed response to my questions!
>
> I think that the authors addressed all my concerns to some degree. However, I have some comments.
>
> 1. Thank you for the clarification! However, I think it is important 1) to outline the assumed behavior of networks (e.g., near-optimal compression by networks), 2) to stress a distinction between the authors' notion of superposition and neuron polysemanticity.
>
> 3. I think there still exists some tension between "observable ground truth", mentioned in Point 2, and the current experiment. Currently, what we see is that in the toy setting, some metrics that should intuitively capture the feature count lead to the same conclusion. It indeed suggests that all metrics capture the same phenomenon, but it still does not constitute a ground truth comparison. (At the same time, I acknowledge that it is not obvious what really constitutes a ground truth for feature counts.)

---

> > ### Author Response · Authors · 2025-09-30
> > **On Assumptions and "Ground Truth"**
> >
> > Thanks for the fast reply! And thank you for the thoughtful comments:
> > 1. We agree with both points.
> > 	- We'll make those assumptions explicit in the revision. What we assume is a) Networks achieve near-optimal compression given their architecture and task constraints (i.e., gradient descent finds reasonably good solutions). b) Networks exhaust lossless compression options before accepting lossy compression with interference. c) The observed feature distribution in trained networks reflects optimal allocation of limited representational resources.
> > 	- We will add a disambiguation box distinguishing polysemanticity and superposition, in particular highlighting this fact, that a network is expected to exhibit polysemanticity even without what we call superposition. This can also help to clarify the results from the dropout experiments.
> > 2. We agree we may need to be more careful in calling this "ground truth validation". The  aspects we show are:
> > 	- **Construct validity**: Our metric tracks the observable transition in interference patterns that we identify with superposition.
> > 	- **Convergent validity**: Multiple metrics converge on the same measurements when applied to toy model weights.
> > 	- **Pipeline robustness**: The measurement remains stable through the SAE extraction pipeline, maintaining high correlation even after the two-stage process (toy model training → SAE extraction). This is critical because in real networks, we can only access features through SAEs, never directly. The toy model's known architecture lets us verify that SAE-based measurement preserves the underlying superposition structure. This validation is impossible in real networks where the "ground truth" feature basis remains unknown.
> >
> > We'll adjust the language in Section 5.1 to say we validate "measurement accuracy in controlled settings", where superposition is directly observable, rather than claiming "ground truth validation."

---

### Review · Reviewer_CPqo · 2025-09-11

**Summary Of Contributions:**

**Summary:** The authors introduce a new metric to quantify "superposition" in neural networks, i.e., the phenomenon in which some neurons encode multiple features in overlapping directions in activation space rather than dedicating each neuron to individual features. Existing methods to quantify this phenomenon have relied on knowledge of the true underlying features in the data and were thus scoped only to toy models. The authors instead propose to train a sparse autoencoder (SAE) on the activations of a model to extract a vocabulary of distinct features and use this to quantify the effective "number" of features using the entropy of the distribution of features extracted by the SAE (the probabilities are calculated over a fixed number of samples). Overall, this is aimed at quantifying the effective information capacity in the activations of a layer and yields a comparative metric that can be used to measure superposition in a relative sense between different conditions. The authors apply this metric to yield several insights, chiefly: (1) showing that dropout increases redundancy when it induces polysemanticity, thus reducing the effective number of features represented given fixed capacity; (2) analysing grokking and detecting it using feature counts; (3) tracking the effective representational capacity across layers in a transformer; and (4) showing that the relationship between adversarial training and the effective number of features depends on task complexity rather than universally causing feature counts to decrease. Overall, the paper aims at providing a better and more widely applicable metric for feature counting and measuring superposition for mechanistic interpretability in neural networks.

**Strengths:**
* The authors have done a good job of motivating their contribution, and the writing and figures are mostly clear (although the paper is quite information-dense).
* The proposed metric is intuitive and principled in its design.
* The paper lists several practical applications and experiments using the metric to analyse the effect of training and design choices on neural network features (in terms of counts and organisation). As such, it moves beyond previous work that was restricted in scope to toy models.

**Weaknesses & Suggestions:**
* The metric relies on the assumption that SAEs can recover the "actual"/"ground truth" features involved in the neural network's computations. This means that the metric is sensitive to how well the SAEs are trained, and I imagine in most practical cases there are not identifiability guarantees for these features. Thus, in practice, the metric would potentially always be computed over several random seeds and hyperparameter configurations for the SAEs and an average value over these configurations would probably be a better estimate to compare. To the authors' credit, they do show in some of their toy experiments that the metric is fairly robust in reasonable hyperparameter ranges.
* However, related to my last point above, it is unclear how expensive computing this metric would be for larger (billion+) parameter models, and how well SAEs would scale to these models. I imagine there would be several more training challenges at that scale and assumptions underlying the metric violated as well.
* Again on a related note, the authors have only tested fairly small models in their experiments, e.g., simple MLPs and CNNs, and Pythia-70M which is a much older and smaller model compared to modern LLMs. I understand that this may be compute-intensive, but truly evaluating the metric's utility for interpretability at scale would require analysing larger models (more modern CNN architectures such as ConvNeXt and ResNeXt, and modern open LLMs that are not too expensive to deploy, such as Llama 2 or Mistral).
* The assumptions underlying the metric, such as feature independence, linear representation assumption, etc. are fairly restrictive and may not hold for most models. However, this is not unique to this paper alone, and I think the authors have done a good job of listing their limitations and have managed to get interesting insights with smaller models despite these limitations.
* One of my key concerns, in general with several works in mechanistic interpretability, is the focus on static layer-wise representations and the lack of modelling dynamics (either through several layers or over several input tokens). Modelling activation dynamics can constrain the features you identify (hence neuroscience's focus on population dynamics [1,2,3]), so it is possible that you might get closer to identifying the actual computation being performed and better feature counts [4]. There are also identifiability guarantees you can get with modelling dynamics [5,6].
* Using the metric in its current form ignores features that largely co-occur or represent the same underlying computation. Instead of calculating the entropy from the raw extracted features directly, the authors could attempt to improve their feature counts by adjusting for "motifs" of features that always co-occur by counting feature pairs across samples. This could help deal with the feature splitting issue.
* Some results such as those in section 6 are quite dense, and could be summarised (maybe at the end) so readers can understand the main insights clearly.

**References:**
1. Vyas, Saurabh et al. "Computation Through Neural Population Dynamics." Annual review of neuroscience vol. 43 (2020): 249-275. doi:10.1146/annurev-neuro-092619-094115
2. Williamson, Ryan C et al. "Bridging large-scale neuronal recordings and large-scale network models using dimensionality reduction." Current opinion in neurobiology vol. 55 (2019): 40-47. doi:10.1016/j.conb.2018.12.009
3. Pandarinath, C et al. "Inferring single-trial neural population dynamics using sequential auto-encoders." Nat Methods 15, 805–815 (2018). doi:10.1038/s41592-018-0109-9
4. Fernando, Jesseba, and Grigori Guitchounts. "Transformer Dynamics: A neuroscientific approach to interpretability of large language models." arXiv preprint arXiv:2502.12131 (2025).
5. Lippe, Phillip, et al. "Citris: Causal identifiability from temporal intervened sequences." International Conference on Machine Learning. PMLR, 2022.
6. Balsells-Rodas, Carles, Yixin Wang, and Yingzhen Li. "On the identifiability of switching dynamical systems." arXiv preprint arXiv:2305.15925 (2023).

**Additional Comments:**

As stated earlier, I work on computational neuroscience and am not an expert in mechanistic interpretability – so my comments might be more high-level and inspired by similar research directions in my field.

For now, I had one additional question: There is a downwards trend in correlation values when scaling the dictionary size beyond 8x the true number of features. Could the authors expand on why this is happening and maybe try to see what happens at, e.g., 64x or beyond?

**Audience:**

Yes

**Audience Explanation:**

This submission focuses on mechanistic interpretability of large models, which is an active and important area of research in the machine learning community. This research has a clear audience among TMLR's readership, i.e., researchers interested in mechanistically understanding the inner workings of deep learning models; it is also clearly aligned with one of the core themes in TMLR's CfP, i.e., "new approaches for analysis, visualization, and understanding of artificial or biological learning systems".

**Broader Impact Concerns:**

I do not see any ethical concerns that could arise uniquely from this contribution. There are several ethical concerns associated with LLM/foundation model research that such methods could help alleviate. However, it is also possible that existing models (or potential future models) violate the foundational assumptions used in this work to extract features from model activations, and I believe these limitations have already been discussed adequately in the submission.

**Claims And Evidence:**

Yes

**Claims Explanation:**

I think the main claim that the proposed metric effectively quantifies superposition has been justified through clear experiments with toy models (Section 5), and also applied to understand feature counts and organisation in several other setups (Section 6). Furthermore, the other main claim that superposition or feature counts may not simply decrease as a result of adversarial training, but change depending on the task the network is performing, has been empirically justified (Section 6.4). Finally, the method's robustness to sensible hyperparameter values/ranges has been demonstrated (Figures 3b & 5a), although I do have one question related to this (see Additional Comments).

**Requested Changes:**

I am not an expert in mechanistic interpretability (my background is in computational neuroscience), so the following points should be viewed more as suggestions to improve the work rather than critical adjustments:

1. Testing the method on models of a larger scale, i.e., more modern and larger CNNs and LLMs.
2. Adjusting feature counts based on features that regularly/always co-occur, thus reducing the effect of feature splitting in SAEs.
3. Incorporating analysis of features across several layers or input tokens, i.e., bringing the method closer to identifying the "dynamics" of computation.
4. Exploring other ways to quantify feature contribution, e.g., the change in outputs or performance by lesioning each feature?

I view the first three points as more important, and especially 1 & 2 as relatively easier to implement in the current framework.

---

> ### Author Response · Authors · 2025-09-25
> **Computational Scalability and Model Architecture**
>
> Thank you for your thoughtful review from a computational neuroscience perspective. Exciting to see the connections between neuroscience and mechanistic interpretability. We address your points below:
>
> ### SAE Sensitivity and Computational Scaling
>
> Regarding computational costs at scale: the expensive component is training the SAE, not computing our metric. Recent work demonstrates SAE scalability to state-of-the-art models—Anthropic successfully trained SAEs on Claude 3 Sonnet [[templeton_scaling_2024]] and OpenAI on GPT-4 [[gao_scaling_2024]], extracting millions of interpretable features. Once trained, our entropy calculation adds negligible overhead. We will clarify that computational bottlenecks lie in SAE training (already demonstrated feasible at scale) rather than our measurement framework. But we agree that, in practice, averaging across multiple seeds and hyperparameter configurations would be best for robust estimates, which may be infeasible for such scale. We suspect though, that more relevant than large numbers, would be the quality of the trained SAE and given a single high-quality SAE the estimate is probably better than with multiple mediocre ones.
>
> ### Model Scale Limitations
>
> We acknowledge testing primarily on smaller models due to computational constraints. While Pythia-70M is indeed modest by current standards, it serves as a proof-of-concept that our framework applies beyond toy models. Scaling to modern architectures (Llama 2, Mistral, ConvNeXt) represents important future work. However, we note that the fundamental question—how do networks organize information under constraints—likely exhibits similar principles across scales. The capacity-complexity tradeoffs we identify should manifest even more clearly in larger models where the abundance/scarcity boundary shifts but doesn't disappear.
>
> ### Dictionary Scaling Correlation Decline
>
> We think the correlation decline beyond 8× dictionary expansion reveals SAE training limitations, related to the feature splitting we investigated with the dedicated dictionary scaling experiment. At extreme scales with fixed L1 regularization (λ=0.1), the dictionary has capacity for fine-grained feature decomposition, but regularization pressure hasn't scaled accordingly.
>
> One solution is scaling regularization with dictionary size—larger dictionaries need stronger L1 pressure to maintain meaningful sparsity. Our dictionary scaling experiments (Figure 5a) show that with appropriate regularization scaling, feature counts stabilize (even though they still grow linearly (much less steep than our exponential increase in dictionary size).

---

> ### Author Response · Authors · 2025-09-25
> **Method and Future Work**
>
> ### Static vs. Dynamic Analysis
>
> Population dynamics in neuroscience reveal computation through temporal evolution—how neural trajectories traverse state space to implement transformations. However, we think superposition is fundamentally a **static property** of representation: how many features can be simultaneously encoded in a fixed-dimensional space.
>
> While tracking features across tokens or layers could reveal computational flow (how features transform during processing), this differs from measuring superposition itself. Superposition concerns the "vocabulary size" available at each layer, not how that vocabulary is used sequentially. We see this time-averaged view isn't a limitation but rather the appropriate level of analysis for this specific phenomenon. That said, understanding how superposition changes during computation (e.g., early layers using dense superposition for feature extraction, late layers using sparse codes for decision-making) remains valuable future work.
>
> ### Feature Co-occurrence and Modularity
>
> This suggestion particularly excites us. Features that consistently co-activate shouldn't be counted independently. We actually attempted a principled solution inspired by quantum entanglement—using the eigenvalue spectrum of the co-occurrence matrix to define an effective feature distribution:
>
> $$\mathbf{C} = \frac{1}{S}\mathbf{Z}\mathbf{Z}^T$$
>
> where eigenvalues would indicate independent feature modules. However, this approach fails at a fundamental level: the covariance matrix rank is bounded by the number of neurons (since SAEs project from $N$ neurons to $D > N$ dictionary elements). This rank constraint makes measuring superposition impossible by construction.
>
> Alternative approaches could include mutual information between feature pairs or hierarchical clustering to group co-activating features. Combining with Matryoshka SAEs might capture relationships across scales. We'll add this to future work, as solving feature splitting would significantly improve accuracy.
>
> ### Feature Lesioning Interpretation
>
> By "lesioning" (ablating) individual SAE features and measuring performance degradation, we could establish causal importance rather than just correlational patterns. However, this introduces complexities: effects might be nonlinear (redundant features compensate) or task-dependent. While valuable for validation, we think this is out of scope but a substantial extension we position as future work.
>
>
> ### Writing Density
>
> We agree our exposition is dense. We will add a summary box highlighting key findings:
> 1. Dropout reduces effective features despite creating polysemantic neurons
> 2. Grokking exhibits sharp feature consolidation at the generalization transition
> 3. Layer-wise feature patterns mirror intrinsic dimensionality studies
> 4. Adversarial training effects depend on capacity×complexity regime
>
> ---
>
> Thank you again for connecting our work to the broader computational neuroscience perspective. Your suggestions about dynamics, co-occurrence, and causal validation highlight extensions also for mechanistic interpretability more generally general.

---

### Review · Reviewer_nNad · 2025-11-03

**Summary Of Contributions:**

The paper defines superposition as lossy compression and proposes a practical metric based on SAE activations: compute a feature‑usage distribution ($p$), set ($F=e^{H(p)}$) as the effective number of features, and measure compression/superposition as ($\psi=F/N$). The metric aligns with toy‑model interference, indicates minimal superposition for compiled algorithmic tasks, shows dropout reduces ($F$) (adds redundancy), captures grokking consolidation, reveals a non‑monotonic layer profile in a small LM, and—most notably—finds that adversarial training can either increase or decrease ($F$) depending on task complexity × model capacity.

Key strengths: principled information‑theoretic metric (effective counts), practicality once SAEs exist, broad demonstrations, improved clarity/statistics in the robustness section.

Key weaknesses: dependence on SAE quality (feature splitting/ghost features), independence/linearity assumptions, small‑scale experiments, and remaining descriptive elements in the adversarial section.

**Audience:**

Yes

**Audience Explanation:**

Readers focused on mechanistic interpretability, representation analysis, and robustness will value a portable metric that ties superposition to effective degrees of freedom and connects interpretability phenomena (dropout, grokking, ID) with a nuanced view of adversarial training (capacity×complexity).

**Broader Impact Concerns:**

Not applicable.

**Claims And Evidence:**

Yes

**Claims Explanation:**

The measurement claims are well supported: toy‑model checks track observable interference; hyperparameter/dictionary studies show stability/plateau; applications (dropout, Tracr, grokking, layer‑wise structure) demonstrate comparative usefulness. The adversarial‑training result—now with formalized hypotheses and meta‑analysis—supports a two‑regime story rather than a universal reduction, while acknowledging non‑monotonic layer behavior. Overall, evidence is sufficient for a measurement contribution.

**Requested Changes:**

1. Foreground the “lossy compression” framing and assumptions: make near‑optimality and lossless‑before‑lossy assumptions explicit; clarify that ($F$) counts virtual neurons (effective degrees), not human‑interpretable features; disambiguate superposition vs. polysemanticity.
2. SAE ablation for measurement stability: on a representative subset (e.g., dropout + adversarial settings), add a comparison with a Gated or Top‑K SAE to show that qualitative ordering of ($F,\psi$) is stable to known ($\ell_1$) issues.
3. Tighten adversarial‑training analysis: integrate the pre‑registered hypotheses and compact stats tables into the main text; report effect sizes/CIs and an ablation over capacity proxies (depth/width) to support the “capacity enables expansion” story. Keep anomalous layer trends visible.
 4. Add a small comparison to effective rank / participation ratio, emphasizing conceptual differences (subspace usage vs. feature compression) and where they agree/diverge.

---

### Decision · Action_Editor_1Y59 · 2025-10-29

**Recommendation:** Accept with minor revision

**Additional Comments:**

The recommendation is based on the reviewers' comments, the action editor's evaluation, and the authors’ response.

This paper investigates the correlation between the information-theoretic measure of sparse autoencoder activations and adversarial robustness. The major concerns of the original paper were centered on the motivation and clarification of the proposed metric. All reviewers find the studied setting novel, and the results provide new insights. The authors’ rebuttal has successfully addressed the major concerns of reviewers. However, the reliability of the sparse autoencoder and the generalization of the claims to larger models beyond the ones studied in this paper remain as follow-up problems to be explored. Overall, I recommend acceptance of this submission with necessary changes. I expect the authors to include the new results and suggested changes during the rebuttal phase to the final version, and fix typos such as missing references.

**Audience:**

Yes

**Audience Explanation:**

Of broad interest to mechanistic interpretability and safety/robustness

**Claims And Evidence:**

Yes

**Claims Explanation:**

The claims of the proposed information-theoretic framework and its connection to adversarial vulnerability are supported by the empirical results.